# SANDWICH BATCH NORMALIZATION

## ABSTRACT

We present Sandwich Batch Normalization (**SaBN**), a frustratingly easy improvement of Batch Normalization (BN) with only a few lines of code changes. SaBN is motivated by addressing the inherent *feature distribution heterogeneity* that one can be identified in many tasks, which can arise from model heterogeneity (dynamic architectures, model conditioning, etc.), or data heterogeneity (multiple input domains). A SaBN factorizes the BN affine layer into one shared *sandwich affine* layer, cascaded by several parallel *independent affine* layers. Its variants include further decomposing the normalization layer into multiple parallel ones, and extending similar ideas to instance normalization. We demonstrate the prevailing effectiveness of SaBN (as well as its variants) as a **drop-in replacement in four tasks**: neural architecture search (NAS), image generation, adversarial training, and style transfer. Leveraging SaBN immediately boosts two state-of-the-art weight-sharing NAS algorithms significantly on NAS-Bench-201; achieves better Inception Score and FID on CIFAR-10 and ImageNet conditional image generation with three state-of-the art GANs; substantially improves the robust and standard accuracy for adversarial defense; and produces superior arbitrary stylized results. We also provide visualizations and analysis to help understand why SaBN works. All our codes and pre-trained models will be released upon acceptance.

## 1 INTRODUCTION

This paper presents a simple, light-weight, and easy-to-implement modification of Batch Normalization (BN) (Ioffe & Szegedy, 2015), yet strongly motivated by various observations (Zając et al., 2019; Deecke et al., 2018; Xie et al., 2019; Xie & Yuille, 2019) drawn from a number of application fields, that *BN has troubles standardizing hidden features with very heterogeneous structures, e.g., from a multi-modal distribution*. We call the phenomenon *feature distribution heterogeneity*. Such heterogeneity of hidden features could arise from multiple causes, often application-dependent:

- One straightforward cause is due to input *data heterogeneity*. For example, when training a deep network on a diverse set of visual domains, that possess significantly different statistics, BN is found to be ineffective at normalizing the activations with only a single mean and variance (Deecke et al., 2018), and often needs to be re-set or adapted (Li et al., 2016).

- Another intrinsic cause could arise from *model heterogeneity*, i.e., when the training is, or could be equivalently viewed as, on a set of different models. For instance, in neural architecture search (NAS) using weight sharing (Liu et al., 2018; Dong & Yang, 2019), training the super-network during the search phase could be considered as training a large set of sub-models (with many overlapped weights) simultaneously. As another example, for conditional image generation (Miyato et al., 2018), the generative model could be treated as a set of category-specific sub-models packed together, one of which would be "activated" by the conditional input each time.

The vanilla BN (Figure 1 (a)) fails to perform well when there is data or model heterogeneity. Recent trends split the affine layer into multiple ones and leverage input signals to modulate or select between them (De Vries et al., 2017; Deecke et al., 2018) (Figure 1 (b)); or even further, utilize several independent BNs to address such disparity (Zając et al., 2019; Xie et al., 2019; Xie & Yuille, 2019; Yu et al., 2018) (Figure 1 (c)). While those relaxations alleviate the data or model heterogeneity, we suggest that they might be *"too loose"* in terms of the normalization or regularization effect.

Let us take adversarial training (AT) (Madry et al., 2017) as a concrete motivating example to illustrate our rationale. AT is by far the most effective approach to improve a deep model's adversarial robustness. The model is trained by a mixture of the original training set ("*clean examples*") and

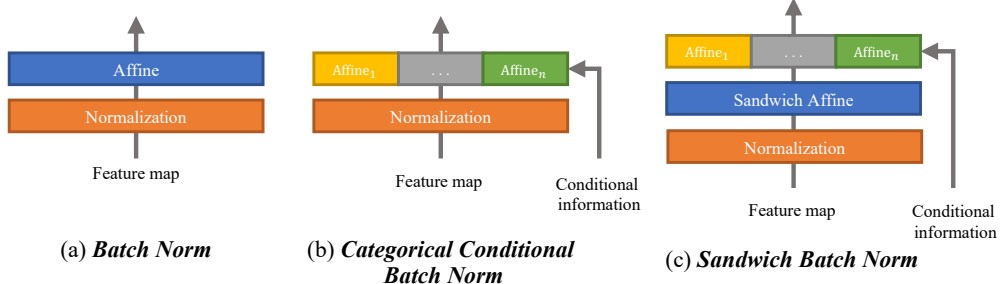

(a) *Batch Norm*  (b) *Categorical Conditional Batch Norm*  (c) *Sandwich Batch Norm*

**Figure 1:** Illustration of (a) the original batch normalization (**BN**), composed of one normalization layer and one affine layer; (b) **Categorical Conditional BN**, composed of one normalization layer following a set of independent affine layers to intake conditional information; (c) our proposed **Sandwich BN**, sequentially composed of one normalization layer, one shared sandwich affine layer, and a a set of independent affine layers.

its attacked counterpart with some small perturbations applied ("*adversarial examples*"). Yet, latest works (Xie et al., 2019; Xie & Yuille, 2019) pointed out that clean and adversarial examples behave like **two different domains** with distinct statistics on the feature level (Li & Li, 2017; Pang et al., 2018). Such data heterogeneity puts vanilla BN in jeopardy for adversarial training, where the two domains are treated as one. (Xie et al., 2019; Xie & Yuille, 2019) demonstrated a helpful remedy to improve AT performance by using two separate BNs for clean and adversarial examples respectively, which allows either BN to learn more stable and noiseless statistics over its own focused domain.

But what may be missing? Unfortunately, using two separate BNs ignores one important fact that the two domains, while being different, are **not totally independent**. Considering that all adversarial images are generated by perturbing clean counterparts only minimally, it is convincing to hypothesize the two domains to be largely overlapped at least (i.e., they still share some hidden features despite the different statistics). To put it simple: while it is oversimplified to normalize the two domains as "same one", it is also unfair and unnecessary to treat them as "disparate two".

More application examples can be found that all share this important structural feature prior, that we (informally) call as "harmony in diversity". For instance, weight-sharing NAS algorithms (Liu et al., 2018; Dong & Yang, 2019; Yu et al., 2018) train a large variety of child models, constituting model heterogeneity; but most child architectures inevitably have many weights in common since they are sampled from the same super net. Similarly, while a conditional GAN (Miyato et al., 2018) has to produce diverse images classes, those classes often share the same resolution and many other dataset-specific characteristics (e.g., the object-centric bias for CIFAR images); that is even more true when the GAN is trained to produce classes of one super-category, e.g., dogs and cats.

**Our Contributions:** Recognizing the need to address feature normalization with "harmony in diversity", we propose a new **SaBN** as illustrated in Fig 1 (c). SaBN modifies BN in a "frustratingly simple" way: it is equipped with two cascaded affine layers: a shared unconditional *sandwich affine* layer, followed by a set of independent affine layers that can be conditioned. Compared to Categorical Conditional BN, the new sandwich affine layer is designed to inject an inductive bias, that all re-scaling transformations will have a shared factor, indicating the commodity. Experiments on the applications of NAS and conditional generation demonstrate that SaBN addresses the *model heterogeneity* issue elegantly, and improves their performance in a plug-and-play fashion.

To better address the *data heterogeneity* altogether, SaBN could further integrate the idea of split/auxiliary BNs (Zając et al., 2019; Xie et al., 2019; Xie & Yuille, 2019; Yu et al., 2018), to decompose the normalization layer into multiple parallel ones. That yields the new variant called **SaAuxBN**. We demonstrate it using the application example of adversarial training. Lastly, we extend the idea of SaBN to Adaptive Instance Normalization (AdaIN) (Huang & Belongie, 2017), and show the resulting **SaAdaIN** to improve arbitrary style transfer.

## 2 RELATED WORK

### 2.1 NORMALIZATION IN DEEP LEARNING

Batch Normalization (BN) (Ioffe & Szegedy, 2015) made critical contributions to training deep convolutional networks and nowadays becomes a cornerstone of the latter for numerous tasks. BN normalizes the input mini-batch of samples by the mean and variance, and then re-scale them with learnable affine parameters. The success of BNs was initially attributed to overcoming internal co-

variate shift (Ioffe & Szegedy, 2015), but later on raises many open discussions on its effect of improving landscape smoothness (Santurkar et al., 2018); enabling larger learning rates (Bjorck et al., 2018) and reducing gradient sensitivity (Arora et al., 2018); preserving the rank of pre-activation weight matrices (Daneshmand et al., 2020); decoupling feature length and direction (Kohler et al., 2018); capturing domain-specific artifacts (Li et al., 2016); reducing BN's dependency on batch sizeIoffe (2017); Singh & Krishnan (2020); Preventing model from elimination singularities Qiao et al. (2019) ; and even characterizing an important portion of network expressivity (Frankle et al., 2020). Inspired by BN, a number of task-specific modifications are proposed by exploiting different normalization axes, such as Instance Normalization (IN) (Ulyanov et al., 2016) for style transfer; Layer Normalization (LN) (Ba et al., 2016) for recurrent networks; Group Normalization (GN) (Wu & He, 2018) for tackling small batch sizes; StochNorm Kou et al. (2020) for fine-tuning task; Passport-aware NormalizationZhang et al. (2020) for model IP protection; and Li et al. (2019a); Wang et al. (2020); Zheng et al. (2020) for image generation.

Several normalization variants have been proposed by modulating BN parameters, mostly the affine layer (mean and variance), to improve the controlling flexibility for more sophisticated usage. For example, Harm *et al.*(De Vries et al., 2017) presents Conditional BN, whose affine parameters are generated as a function of the input. Similarly, Conditional IN (Dumoulin et al., 2016) assigns each style with independent IN affine parameters. In (Miyato et al., 2018), the authors developed Categorical Conditional BN for conditional GAN image generation, where each generated class has its independent affine parameters. Huang & Belongie (Huang & Belongie, 2017) presented Adaptive IN (AdaIN), which used the mean and variance of style image to replace the original affine parameter, achieving arbitrary style transfer. Spatial adaptivity (Park et al., 2019) and channel attention (Li et al., 2019b) managed to modulate BN with higher complexities.

A few latest works investigate to use multiple normalization layer instead of one in BN. (Deecke et al., 2018) developed mode normalization by employing a mixture-of-experts to separate incoming data into several modes and separately normalizing each mode. (Zając et al., 2019) used two separate BNs to address the domain shift between labeled and unlabeled data in semi-supervised learning. Very recently, (Xie & Yuille, 2019; Xie et al., 2019) reveal the two-domain issue in adversarial training and find improvements by using two separate BNs (AuxBN).

## 2.2 BRIEF BACKGROUNDS FOR RELATED APPLICATIONS

We leverage four important applications as test beds. All of them appear to be oversimplified by using the vanilla BN, where the feature homogeneity and heterogeneity are not properly handled. We briefly introduce them below, and will concretely illustrate where the heterogeneity comes from and how our methods resolve the bottlenecks in Sec. 3.

**Generative Adversarial Network** Generative adversarial networks (GANs) have been prevailing since its origin (Goodfellow et al., 2014a) for image generation. Many efforts have been made to improve GANs, such as modifying loss function (Arjovsky et al., 2017; Gulrajani et al., 2017; Jolicoeur-Martineau, 2018), improving network architecture (Zhang et al., 2018; Karras et al., 2019; Gong et al., 2019) and adjusting training procedure (Karras et al., 2017). Recent works also tried to improve the generated image quality by proposing new normalization modules, such as Categorical Conditional BN and spectral normalization (Miyato et al., 2018).

**Neural Architecture Search (NAS)** The goal of NAS is to automatically search for an optimal model architecture for the given task and dataset. It was first proposed in (Zoph & Le, 2016) where a reinforcement learning algorithm iteratively samples, trains and evaluates candidate models from the search space. Due to its prohibitive time cost, weight-sharing mechanism was introduced (Pham et al., 2018) and becomes a popular strategy to accelerate the training of sampled models (Liu et al., 2018). However, weight-sharing causes performance deterioration due to unfair training (Chu et al., 2019). In addition, a few NAS benchmarks (Ying et al., 2019; Dong & Yang, 2020; Zela et al., 2020) were recently released, with ground-truth accuracy for candidate models pre-recorded, enabling researchers to evaluate the performance of search method more easily.

**Adversarial Robustness** Deep networks are notorious for the vulnerability to adversarial attacks (Goodfellow et al., 2014b). In order to enhance adversarial robustness, countless training approaches have been proposed. (Dhillon et al., 2018; Papernot & McDaniel, 2017; Xu et al., 2017; Meng & Chen, 2017; Liao et al., 2018; Madry et al., 2017). Among them, adversarial training (AT) (Madry et al., 2017) are arguably the strongest, which train the model over a mixture of clean and perturbed data. Overall, the normalization in AT has, to our best knowledge, not been studied in depth. A

pioneer work (Xie et al., 2019) introduce an auxiliary batch norm (AuxBN) to improve the clean image recognition accuracy.

**Neural Style Transfer** Style transfer is a technique generating a stylized image, by combining the content of one image with the style of another. Various improvements are made on the normalization methods in this area. Ulyanov et al. (2016) proposed Instance Normalization, improving the stylized quality of generated images. Conditional Instance Normalization (Dumoulin et al., 2016) and Adaptive Instance Normalization (Huang & Belongie, 2017) are proposed, enabling the network to perform arbitrary style transfer.

## 3 SANDWICH BATCH NORMALIZATION

**Formulation:** Given the input feature $\mathbf{x} \in \mathbb{R}^{N \times C \times H \times W}$ ($N$ denotes the batch size, $C$ the number of channels, $H$ the height, and $W$ the width), the vanilla Batch Normalization (BN) works as:

$$\mathbf{h} = \boldsymbol{\gamma}(\frac{\mathbf{x} - \mu(\mathbf{x})}{\sigma(\mathbf{x})}) + \boldsymbol{\beta}, \tag{1}$$

where $\mu(\mathbf{x})$ and $\sigma(\mathbf{x})$ are the running estimates (or batch statistics) of input $x$'s mean and variance along the $(N, H, W)$ dimension. $\boldsymbol{\gamma}$ and $\boldsymbol{\beta}$ are the learnable parameters of the affine layer, and both are of shape $C$. However, as the vanilla BN only has a single re-scaling transform, it will simply treat any latent heterogeneous features as a single distribution. As an improved variant, Categorical Conditional BN (CCBN) (Miyato et al., 2018) is proposed to remedy the heterogeneity issue in the task of conditional image generation, boosting the quality of generated images. Categorical Conditional BN has a set of independent affine layers, whose activation is conditioned by the input domain index. It can be expressed as:

$$\mathbf{h} = \boldsymbol{\gamma}_i(\frac{\mathbf{x} - \mu(\mathbf{x})}{\sigma(\mathbf{x})}) + \boldsymbol{\beta}_i, i = 1, ..., C, \tag{2}$$

where $\boldsymbol{\gamma}_i$ and $\boldsymbol{\beta}_i$ are parameters of the $i$-th affine layer. Concretely, $i$ is the expected output class in the image generation task (Miyato & Koyama, 2018). However, we argue that this "separate/split" modification might be "too loose", ignoring the fact that the distributions of the expected generated images overlap largely (due to similar texture, appearance, illumination, object location, scene layout, etc.). Hence to better handle both the latent homogeneity and heterogeneity in $\mathbf{x}$, we present Sandwich Batch Normalization (**SaBN**), that is equipped with both a shared sandwich affine layer and a set of independent affine layers. **SaBN can be concisely formulated as**:

$$\mathbf{h} = \boldsymbol{\gamma}_i(\boldsymbol{\gamma}_{sa}(\frac{\mathbf{x} - \mu(\mathbf{x})}{\sigma(\mathbf{x})}) + \boldsymbol{\beta}_{sa}) + \boldsymbol{\beta}_i, i = 1, ..., C. \tag{3}$$

As depicted in Fig. 1 (d), $\boldsymbol{\gamma}_{sa}$ and $\boldsymbol{\beta}_{sa}$ denote the new sandwich affine layer, while $\boldsymbol{\gamma}_i$ and $\boldsymbol{\beta}_i$ are the $i$-th affine parameters, conditioned on categorical inputs. Implementation-wise, SaBN only takes **a few lines of code changes** compared to vanilla BN: please see **appendix** Fig. 6 for pseudo codes.

### 3.1 UNIFYING HOMOGENEITY & HETEROGENEITY FOR CONDITIONAL IMAGE GENERATION

As one of the state-of-the-art GANs, SNGAN (Miyato et al., 2018) successfully generate high quality images in conditional image generation tasks with Categorical Conditional BN (CCBN). Intuitively, it uses independent affine layers to disentangle the image generation of different classes. Thus the generative model could be treated as a set of category-specific sub-models in one model, one of which would be "activated"

**Table 1:** The best Inception Scores ("IS", ↑) and FIDs (↓) achieved by conditional SNGAN, BigGAN, and AutoGAN-top1, using CCBN and SaBN on CIFAR-10 and ImageNet (dogs & cats).

| Model | CIFAR-10 | | ImageNet (dogs & cats) | |
|---|---|---|---|---|
| | IS | FID | IS | FID |
| AutoGAN-top1 | 8.43 | 10.51 | - | - |
| BigGAN | 8.91 | 8.57 | - | - |
| SNGAN | 8.76 | 10.18 | 16.75 | 79.14 |
| AutoGAN-top1-SaBN | 8.72(+0.29) | 9.11(−1.40) | - | - |
| BigGAN-SaBN | 9.01(+0.10) | 8.03(−0.54) | - | - |
| SNGAN-SaBN | 8.89(+0.13) | 8.97(−1.21) | 18.31(+1.56) | 60.38(−18.76) |

by the conditional class input each time, incurring the **model heterogeneity**. However, only leveraging separate affine layers ignores the fact that the expected generated domains are not totally independent with each other. These classes share the same image resolution, similar illumination, object location, and some intrinsic dataset-specific characteristics, which suggest strong **homogeneity**.

This motivates us to unify both homogeneity and heterogeneity for conditional image generation with SaBN. We choose three representative GAN models, SNGAN, BigGAN (Brock et al., 2018) and AutoGAN-top1 (Gong et al., 2019), as our backbones. SNGAN and BigGAN are equipped with Categorical Conditional BN. AutoGAN-top1 originally has no normalization layer and was designed for unconditional image

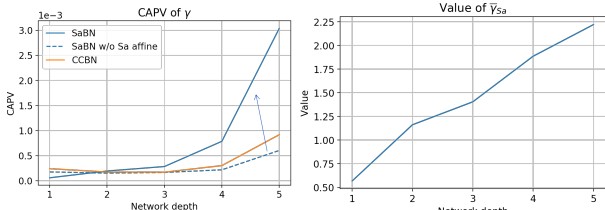

**Figure 2:** The CAPV value of $\gamma$ (left) and the shared sandwich parameter $\gamma_{Sa}$'s value (right) along the network depth.

generation, we manually insert Categorical Conditional BN into its generator to adapt it to the conditional image generation task. We then construct SNGAN-SaBN, BigGAN-SaBN and AutoGAN-SaBN, by replacing all Categorical Conditional BN in the above baselines with our SaBN.

We test all the above models on CIFAR-10 dataset (Krizhevsky et al., 2009) (10 categories, resolution $32 \times 32$). Furthermore, we test SNGAN and SNGAN-SaBN on high resolution image generation task on ImageNet (Deng et al., 2009), using the subset of all 143 classes belonging to the dog and cat super-classes, cropped to resolution $128 \times 128$) following (Miyato et al., 2018)'s setting. Inception Score (Salimans et al., 2016) (the higher the better) and FID (Heusel et al., 2017) (the lower the better) are adopted as evaluation metrics. We summarize the best performance the models has achieved during training into Table 1. We find that SaBN can consistently improve the generative quality of all three baseline GAN models. We also visualize the image generation results of all compared GANs in Fig. 21, 22, 23 and 24 at appendix.

**Understanding SaBN by Visualization.** One might be curious about the effectiveness of SaBN, since at the inference time, the shared sandwich affine layer can be multiplied/merged into the independent affine layers, making the inference form of SaBN completely the same as Categorical Conditional BN. Hence to better understand how SaBN benefits the conditional image generation task, we dive into the inductive role played by the shared sandwich affine parameter.

We choose SNGAN and SNGAN-SaBN on ImageNet as our testbed. Specifically, we propose a new measurement called *Class-wise Affine Parameters Variance* (**CAPV**), which aims to indicate how much class-specific heterogeneity is introduced with the set of independent affine parameters. For Categorical Conditional BN (CCBN), we define its CAPV for $\gamma$ as $V_{\text{CCBN}}(\gamma) = \text{Var}([\bar{\gamma}_1, \bar{\gamma}_2, \cdots, \bar{\gamma}_C])$, where $\bar{\gamma}_i$ represents the channel-wise average value of $\gamma_i$. Similarly, the CAPV of SaBN is defined as $V_{\text{SaBN}}(\gamma) = \text{Var}([\overline{\gamma_{Sa} \cdot \gamma_1}, \overline{\gamma_{Sa} \cdot \gamma_2}, \cdots, \overline{\gamma_{Sa} \cdot \gamma_C}])$. A larger CAPV value implies more heterogeneity.

We plot the CAPV values for both SNGAN and SNGAN-SaBN at each layer in Fig. 2 (left). The solid

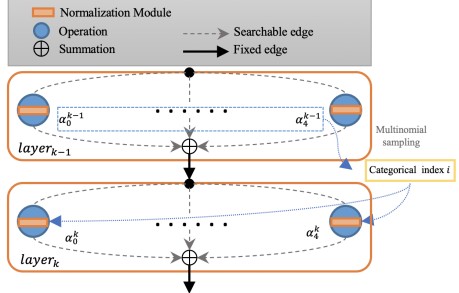

**Figure 3:** We depict two consecutive layers in the super-network. By default, a BN is integrated into each operation in vanilla DARTS, except Zero and Skip-connection operation. The output of each layer is the sum of all operation paths' output, weighted by their associated architecture parameter $\alpha$. **Model heterogeneity** is introduced during the summation process, due to the difference among each parallel operations. In the meanwhile, different operations still maintain an **intrinsic homogeneity** and are not completely independent, as they all share the same original input, and their gradient is also estimated from the same loss function.

blue line (SaBN) is lower than orange line (CCBN) in the shallow layers, but soon surpasses the latter at the deeper layers. We additionally plot a dashed blue line, obtained by removing the shared sandwich affine layer from the SaBN parameters (i.e., only independent affine). Their comparison indicates that the shared sandwich affine layer helps inject an inductive bias: compared to using Categorical Conditional BN, training with SaBN will enforce to shallow layers to preserve more feature homogeneity, while encouraging the deeper layer to introduce more class-specific feature heterogeneity. In other plain words, SaBN seems to have shallower and deeper layers more "focused" on each dedicated role (common versus class-specific feature extractors). We also plot the value of shared sandwich affine parameters $\gamma_{Sa}$ (averaged across channel) along the network depth, in the right figure in Fig.2. It is also aligned with our above observation.

## 3.2 ARCHITECTURE HETEROGENEITY IN NEURAL ARCHITECTURE SEARCH (NAS)

Recent NAS works formulate the search space as a super-network which contains all candidate operations and architectures, and the goal is to find a sub-network of a single path that of the optimal performance. To support the search over the architectures in the super-network, DARTS (Liu et al., 2018) assigns each candidate operation a trainable parameter $\alpha$, and the search problem is solved by alternatively optimizing the architecture parameters $\alpha$ and the model weights via stochastic gradient descent. The architecture parameters $\alpha$ can be treated as the magnitude of each operation, which will help rank the best candidates after the search phase.

In Fig. 3, we concretely illustrate the origin of the **model heterogeneity** in the supernet. To disentangle the mixed model heterogeneity during search process and still maintain the intrinsic homogeneity, we replace the BN in each operation path with a SaBN in the second layer (same for the first layer if it is also downstream of layers ahead). Ideally, the number of independent affine layers shall be set to the total number of unique architectures in the search space, enabling the architecture-wise disentanglement. However, it would be impractical as the search space size (number of unique architecture) is

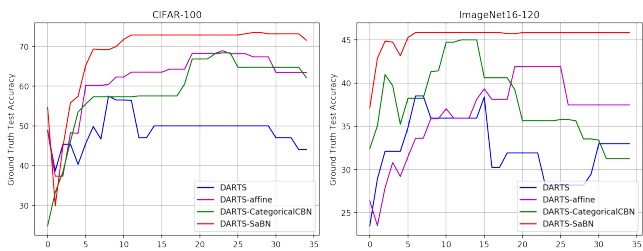

**Figure 4:** Results of architecture search on CIFAR-100 and ImageNet16-120, based on DARTS. At the end of each searching epoch, the architecture is derived from current $\alpha$ values. The x-axis is the searching epoch. The y-axis is the ground truth test accuracy of current epoch's architecture, obtained via querying NAS-Bench-201. Each experiment is run for three times with different random seeds. Each curve in the figure is averaged across them.

usually larger than $10^4$. Thus we adopt a greedy way that we only consider to disentangle previous layer's architecture (operations). Specifically, the number of independent affine layers in the SaBN equals to the total number of candidate operation paths of the connected previous layer. The categorical index $i$ of SaBN during searching is obtained by applying a multinomial sampling on the softmax of previous layers' architecture parameters: $\text{softmax}([\alpha_0, \alpha_1, \alpha_2, ...\alpha_{n-1}])$.

To validate the efficacy of our SaBN, we conduct ablation study of four settings: 1) **no affine** (i.e., $\gamma = 1, \beta = 0$, as in vanilla DARTS), where by default the learning of affine parameters of BN in each operation path is disabled (Liu et al., 2018); 2) **homogeneity only** ("DARTS-affine"), where the learning of affine parameters of BN are enabled in each operation path in "DARTS"; 3) **heterogeneity only** ("DARTS-CategoricalCBN"), where the BN in each operation path of "DARTS" are replaced by the Categorical Conditional BN (Miyato et al., 2018); 4) **homogeneity and heterogeneity** ("DARTS-SaBN"), where all operation paths' BNs in "DARTS" are replaced by SaBN. Following the suggestion by Dong & Yang (2020), all experiments use batch statistics instead of keeping running estimates of the mean and variance in the normalization layer.

We conduct our experiments on CIFAR-100 (Krizhevsky et al., 2009) and ImageNet16-120 (Chrabaszcz et al., 2017) using NAS-Bench-201 (Dong & Yang, 2020) (Fig. 4). Early stopping is applied for the searching phase as suggested in (Liang et al., 2019). We can observe that SaBN dominates on both CIFAR-100 and ImageNet cases. Surprisingly, we also notice that by simply

**Table 2:** The searched results top-1 accuracy of the four methods on NAS-Bench-201. Our proposed approach achieves the highest accuracy, with the lowest standard deviation.

| Method | CIFAR-100 | ImageNet |
|---|---|---|
| DARTS | $44.05 \pm 7.47$ | $36.47 \pm 7.06$ |
| DARTS-affine | $63.46 \pm 2.41$ | $37.26 \pm 7.65$ |
| DARTS-CCBN | $62.16 \pm 2.62$ | $31.25 \pm 6.20$ |
| DARTS-SaBN (ours) | $\mathbf{71.56 \pm 1.39}$ | $\mathbf{45.85 \pm 0.72}$ |

turning on the affine in the original DARTS, DARTS-affine gains fairly strong improvements. The performance gap between DARTS-affine and DARTS-SaBN demonstrate the effectiveness of the independent affine layers in SaBN.

Experiments also shows that CCBN does help improve search performance. However, it falls largely behind SaBN, indicating the shared sandwich affine layer to also be vital. In Fig. 15 at appendix, we can observe the shared sandwich affine layer helps to preserve more homogeneity. The ground-truth accuracy of the final searched architecture is summarized in Tab. 2. Besides, we also find SaBN

works well on another weight-sharing search method, GDAS (Dong & Yang, 2020). The results are shown in Sec. A.2.3 at appendix.

## 3.3 SANDWICH AUXILIARY BATCH NORM IN ADVERSARIAL ROBUSTNESS

AdvProp (Xie et al., 2019) successfully utilized adversarial examples to boost network Standard Testing Accuracy (SA) by introducing Auxiliary Batch Norm (AuxBN). The design is quite simple: an additional BN is added in parallel of the original BN, where the original BN (clean branch) takes the clean image as input, while the additional BN (adversarial branch) is fed with only adversarial examples during training. That intuitively disentangles the mixed clean and adversarial distribution (**data heterogeneity**) into two splits, guaranteeing the normalization statistics and re-scaling are exclusively performed in either domain.

However, one thing missed is that the domains of clean and adversarial images overlap largely, as adversarial images are generated by perturbing clean counterparts minimally. This inspires us to present a novel **SaAuxBN**, by leveraging domain-specific normalizations and affine layers, and also a shared sandwich affine layer for homogeneity preserving, SaAuxBN can be defined as:

$$\mathbf{h} = \boldsymbol{\gamma}_i(\boldsymbol{\gamma}_{sa}(\frac{\mathbf{x} - \mu_i(\mathbf{x})}{\sigma_i(\mathbf{x})}) + \boldsymbol{\beta}_{sa}) + \boldsymbol{\beta}_i, i = 0, 1. \tag{4}$$

$\mu_i(\mathbf{x})$ and $\sigma_i(\mathbf{x})$ denote the $i$-th (moving) mean and variance of input, where $i = 0$ for adversarial images and $i = 1$ for clean images. We use independent normalization layer to decouple the data from two different distributions, i.e., the clean and adversarial.

For a fair comparison, we follow the settings in Madry et al. (2017). In the adversarial training, we adopt $\ell_\infty$ based 10 steps Projected Gradient Descent (PGD) (Madry et al., 2017) with step size $\alpha = \frac{2}{255}$ and maximum perturbation magnitude $\epsilon = \frac{8}{255}$; As for assessing RA, PGD-20 with the same configuration is adopted.

We replace AuxBN with SaAuxBN in AdvProp (Xie et al., 2019) and find it can further improve SA of the network with its clean branch. The experiments are conducted on CIFAR-10 (Krizhevsky et al., 2009) with ResNet-18 (He et al., 2016) backbone, and the results are presented in Tab. 3.

**Table 3:** Performance (SA) of different BN settings on clean branch.

| Evaluation | BN | AuxBN (clean branch) | ModeNorm | SaAuxBN (clean branch) |
|---|---|---|---|---|
| Clean (SA) | 87.25 | 94.38 | 86.10 | **94.47** |

We further conduct an experiment to test the SA and Robust Testing Accuracy (RA) of the network using the adversarial branch of AuxBN and SaAuxBN. The comparison results are presented in Tab. 4.

**Table 4:** Performance (SA&RA) of different BN settings. During evaluation, only the adversarial path is activated in AuxBN and SaAuxBN.

| Evaluation | BN | AuxBN (adv branch) | ModeNorm | SaAuxBN (adv branch) |
|---|---|---|---|---|
| Clean (SA) | **87.25** | 85.08 | 86.10 | 87.07 |
| PGD-10 (RA) | 42.82 | 43.30 | 44.60 | **45.72** |
| PGD-20 (RA) | 40.84 | 41.98 | 42.81 | **44.23** |

Tab. 4 shows that BN still achieves the highest performance on SA, but falls a lot on RA compared with other methods. Our proposed SaAuxBN is on par with the vanilla BN in terms of SA, while has significantly better results on RA than any other approaches. Compared with SaAuxBN, AuxBN suffers from worse SA and RA, indicating that both the shared sandwich affine layer are key to the disentanglement of adversarial domain from the clean domain, with their shared statistics properly preserved. The behaviors of AuxBN and SaAuxBN are visualized in Fig. 20 at appendix, which suggests that the sandwich affine layer here mainly encourages to enhance feature homogeneity.

We additionally include ModeNorm (Deecke et al., 2018) as an ablation in our experiments, which was proposed to deal with multi-modal distributions inputs, i.e., data heterogeneity. It shares some similarity with AuxBN as both consider multiple independent norms. ModeNorm achieves fair

performance on both SA and RA, while still lower than SaAuxBN. The reason might be the output of ModeNorm is a summation of two features weighted by a set of learned gating functions, which still mixes the statistics from two domains, leading to inferior performance in the attack scenario.

### 3.4 STYLE TRANSFER WITH SANDWICH ADAPTIVE INSTANCE NORMALIZATION

Huang & Belongie (Huang & Belongie, 2017) achieves arbitrary style transfer by introducing Adaptive Instance Norm (AdaIN). The AdaIN framework is composed of three parts: Encoder, AdaIN and Decoder. Firstly, the Encoder will extract content feature and style feature from content and style images. Then the AdaIN is leveraged to perform style transfer on feature space, producing a stylized content feature. The Decoder is learnt to decode the stylized content feature to stylized images. This framework is trained end-to-end with two loss terms, a content loss and a style loss.

Concretely, AdaIN firstly performs a normalization on the content feature, then re-scale the normalized content feature with style feature's statistic. It can be formulated as:

$$\mathbf{h} = \sigma(\mathbf{y})(\frac{\mathbf{x} - \mu(\mathbf{x})}{\sigma(\mathbf{x})}) + \mu(\mathbf{y}), \tag{5}$$

where $y$ is the style input, $x$ is the content input. Note that $\mu$ and $\sigma$ here are quite different from BN, which are performed along the spatial axes $(H, W)$ for each sample and each channel. Obviously, style-dependent re-scale may be too loose and might further amplify the intrinsic **data heterogeneity** brought by the variety of the input content images, undermining the network's ability of maintaining the content information in the output. In order to reduce the **data heterogeneity**, we propose to insert a shared sandwich affine layer after the normalization, which introduce **homogeneity** for the style-dependent re-scaling transformation. Hereby, we present SaAdaIN:

$$\mathbf{h} = \sigma(\mathbf{y})(\boldsymbol{\gamma}_{sa}(\frac{\mathbf{x} - \mu(\mathbf{x})}{\sigma(\mathbf{x})}) + \boldsymbol{\beta}_{sa}) + \mu(\mathbf{y}), \tag{6}$$

Besides AdaIN, we also include Instance-Level Meta Normalization with Instance Norm (ILM+IN) proposed by Jia et al. (2019) as a task-specific comparison baseline. Its style-independent affine is not only conditioned on style information, but also controlled by the input feature. Our training settings for all models are kept identical with (Huang & Belongie, 2017). We depict the loss curves of the training process in Fig. 5.

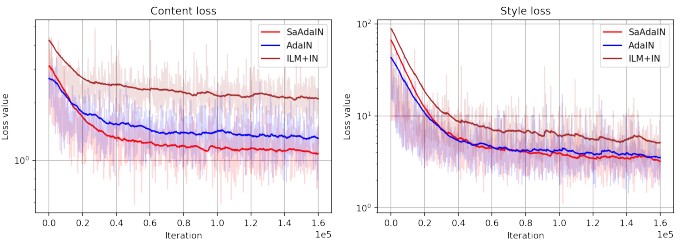

**Figure 5:** The content loss and the style loss of using AdaIN, ILM+IN and SaAdaIN. The noisy shallow-color curves are the original data. The foreground smoothed curves are obtained via applying exponential moving average on the original data.

We can notice that both the content loss and style loss of the proposed SaAdaIN is lower than that of AdaIN and ILM+IN. This observation demonstrates that the shared sandwich affine layer in our SaAdaIN is beneficial for the network to preserve the semantic information of the original input content, while also better migrating and merging style information.

Furthermore, the qualitative visual results are demonstrated in Fig. 26 at appendix (Better zoomed in and viewed in color). The leftmost column displays content images and referenced style images. The next three columns are the stylized outputs using AdaIN, ILM+IN and SaAdaIN, respectively.

## 4 CONCLUSION

We present SaBN and its variants as plug-and-play normalization modules, which are motivated by addressing model & data heterogeneity issues. We demonstrate their effectiveness on several tasks, including neural architecture search, adversarial robustness, conditional image generation and arbitrary style transfer. Our future work plans to investigate the performance of SaBN on more applications, such as semi-supervised learning (Zając et al., 2019), slimmable models, and once-for-all training (Yu et al., 2018).

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

# A APPENDIX

## A.1 IMPLEMENTATION DETAILS

```python
def BatchNorm(x, gamma, beta, running_mean, running_var, eps=1e-5,
    momentum=0.1):
    # x: input features with shape [N,C,H,W]
    # gamma, beta: scale factors with shape [1,C,1,1]
    mean, var = tf.nn.moments(x, [0, 2, 3], keep dims = True)

    running_mean = (1 - momentum) * running_mean + momentum * mean
    running_var = (1 - momentum) * running_var + momentum * var
    x = (x - running_mean) / tf.sqrt(running_var + eps)
    return x * gamma + beta

def SaBatchNorm(x, sa_gamma, sa_beta, gammas, betas, index,
    running_mean, running_var, eps=1e-5, momentum=0.1):
    # x: input features with shape [N,C,H,W]
    # sa_gamma, sa_beta: shared scale factor with shape [1,C,1,1]
    # gammas, betas: a list of scale factors with shape [1,C,1,1]
    mean, var = tf.nn.moments(x, [0, 2, 3], keep_dims = True)
    running_mean = (1 - momentum) * running_mean + momentum * mean
    running_var = (1 - momentum) * running_var + momentum * var

    x = (x - running_mean) / tf.sqrt(running_var + eps)
    x = x * sa_gamma + sa_beta
    return x * gammas[index] + betas[index]

def SaAuxBatchNorm(x, sa_gamma, sa_beta, gammas, betas, index,
    running_means, running_vars, eps=1e-5, momentum=0.1):
    # x: input features with shape [N,C,H,W]
    # sa_gamma, sa_beta: shared scale factors with shape [1,C,1,1]
    # gammas, betas: a list of scale factors with shape [1,C,1,1]
    mean, var = tf.nn.moments(x, [0, 2, 3], keep dims = True)
    running_means[index] = (1 - momentum) * running_means[index]
                            + momentum * mean
    running_vars[index] = (1 - momentum) * running_vars[index]
                           + momentum * var

    x = (x - running_means[index]) / tf.sqrt(running_vars[index]
        + eps)
    x = x * sa_gamma + sa_beta
    return x * gammas[index] + betas[index]
```

**Figure 6:** Pseudo Python code of BN, SaBN and SaAuxBN with TensorFlow. We highlight the main difference between our approaches with vanilla BN.

## A.2 ADDITIONAL RESULTS IN NAS

### A.2.1 SEARCH SPACE OF NAS-BENCH-201

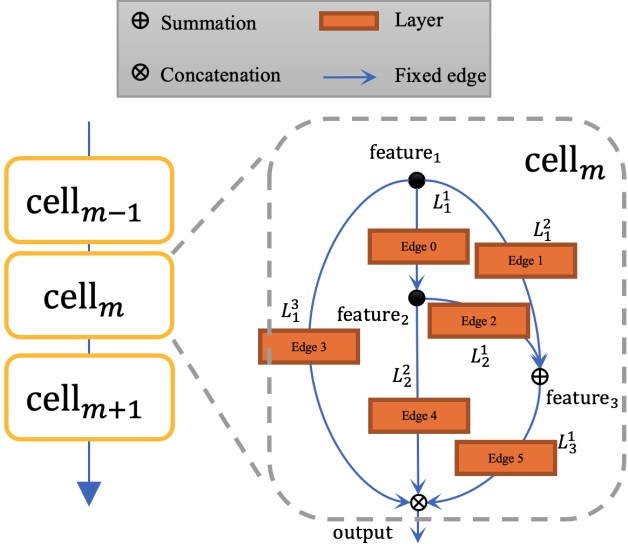

**Figure 7:** The search space of NAS-Bench-201.

Our experiments of NAS are conducted on NAS-Bench-201, which adopts a cell-based search space, shown in Fig. 7. The whole search space is composed by stacking several cells. Each cell consists of several layers $L_j^k$, which is composed of several parallel operation paths as illustrated in Fig. 2 in our paper. $L_j^k$ denotes the $k$-th layer which take feature$_j$ as input. In Sec. 3.1 of our paper, we have concretely illustrate the detailed implementation of the SaBN under the circumstance a layer only has one connected previous layer. However, we can observe there is layer having multiple connected previous layers. Take $L_3^1$ as an example, it is connected with $L_1^2$ and $L_2^1$. In this case, the number of independent affine parameter in SaBN at $L_3^1$ would be $n^2$, where $n$ is the number of parallel operation paths in each layer.

### A.2.2 THE ADDITIONAL RESULTS OF SABN IN DARTS

**Visualization of architecture parameters in DARTS**

We visualize the searching curves of the architecture parameters on each edge of the cell, in Fig. 8, 9, 10, for DARTS, DARTS-CCBN and DARTS-SaBN respectively. In the case of DARTS (Fig. 8), we can see that "skip_connect" dominates on most of edges during the search, yielding bad architectures when search ends. DARTS-CCBN is in favor of both "skip_connect" and "none" operations at last, in Fig. 9. For our DARTS-SaBN in Fig. 10, the weight of "skip_connect" in architecture parameters climbs at first, but drops immediately after a few epochs, whereas the "nor_conv_3x3" takes the lead in most of edges.

**Visualization of discovered architectures in DARTS**

We visualize the discovered architectures of DARTS, DARTS-CCBN and DARTS-SaBN in Fig. 11. It shows the architecture that discovered by DARTS is fully dominated by "skip_connect". We further analyze the operator composition of searched architectures of DARTS and DARTS-SaBN in Fig. 12, based on their final searched architecture parameters. We can clearly see that "skip_connect" is highly preferred by DARTS, where DARTS-SaBN favors "nor_conv_3x3".

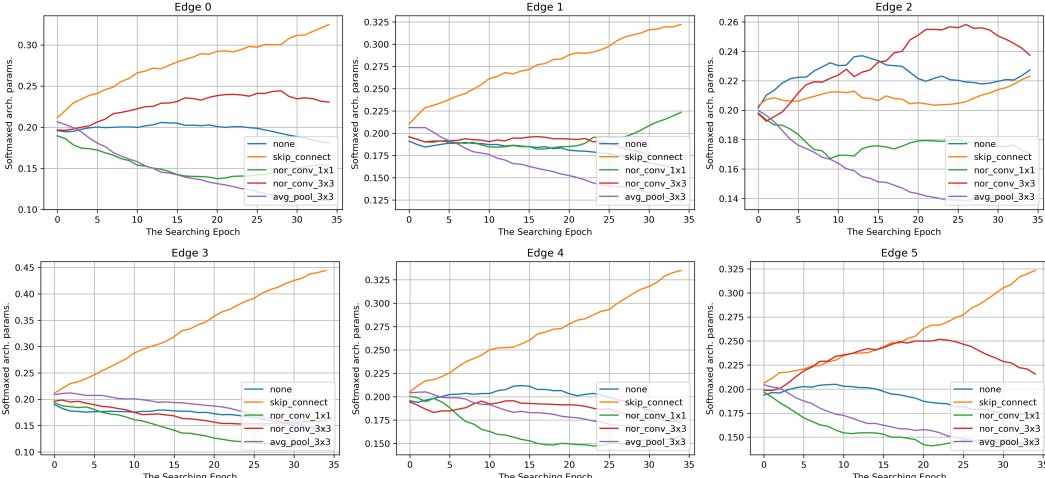

**Figure 8:** Architecture parameters (after softmax) on each edge in DARTS.

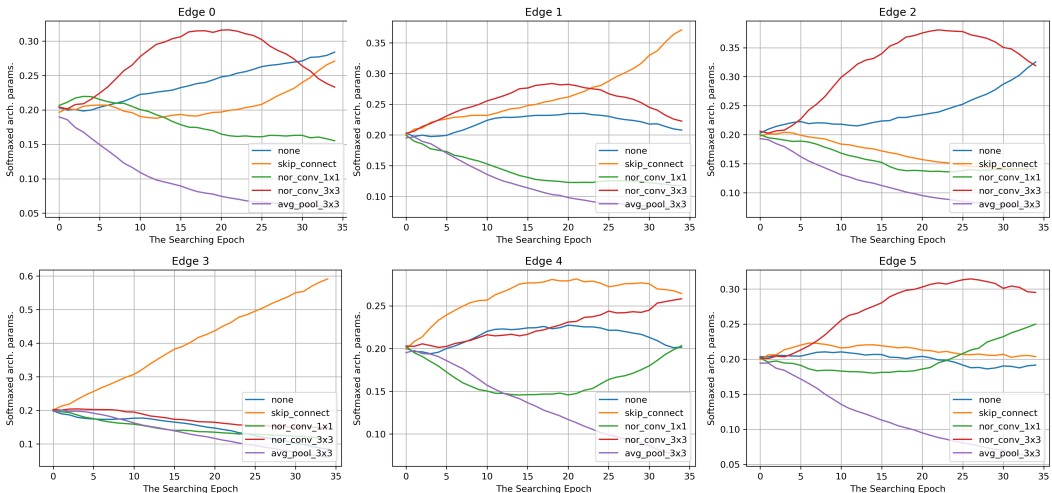

**Figure 9:** Architecture parameters (after softmax) on each edge in DARTS-CCBN.

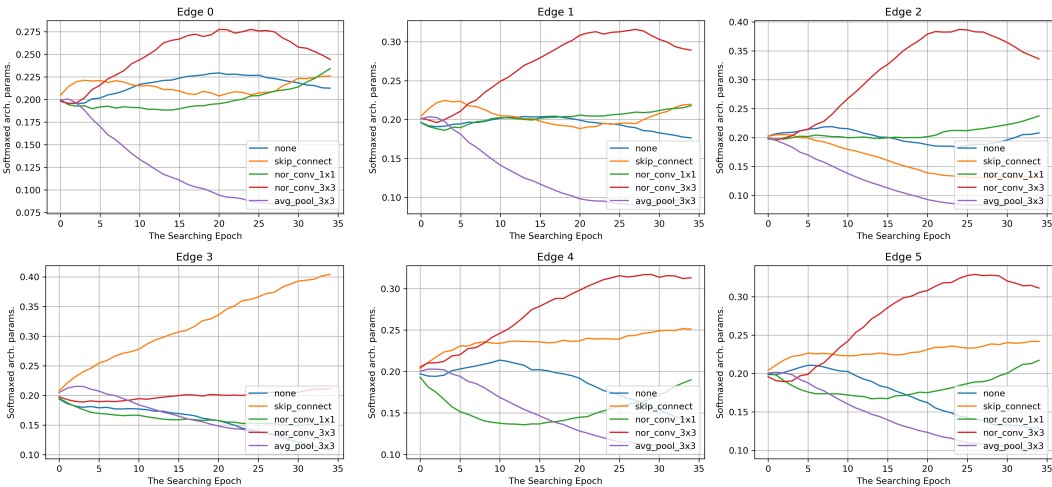

**Figure 10:** Architecture parameters (after softmax) on each edge in our DARTS-SaBN.

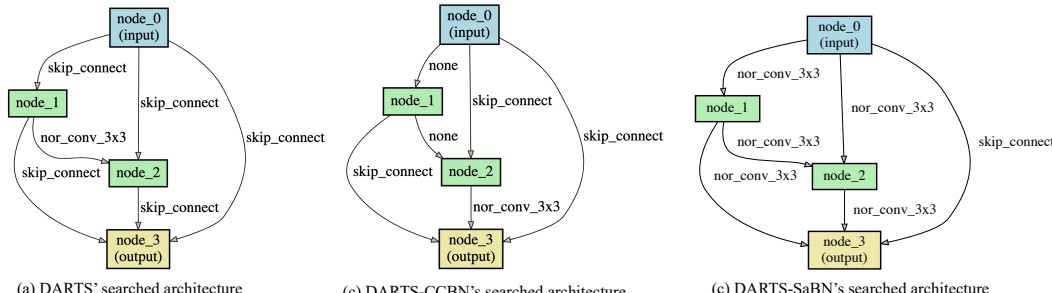

(a) DARTS' searched architecture    (c) DARTS-CCBN's searched architecture    (c) DARTS-SaBN's searched architecture

**Figure 11:** The architectures searched by DARTS are dominated by "skip_connect" and the architecture of DARTS-CCBN is full of both "skip_connect" and "none". In contrast, DARTS-SaBN highly prefers "nor_conv_3x3".

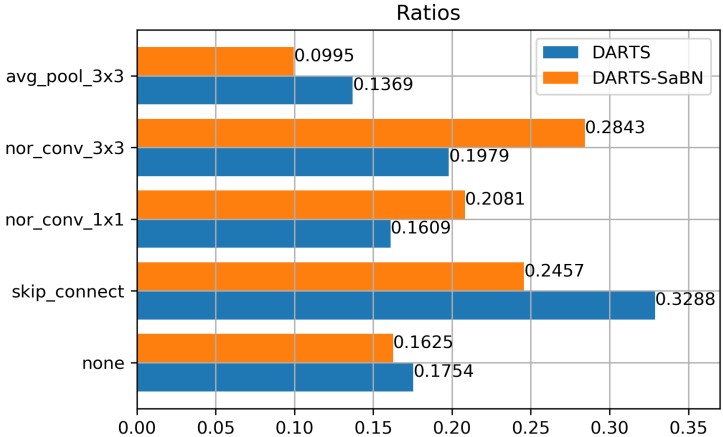

**Figure 12:** The operation statistics of the searched architecture from DARTS and DARTS-SaBN.

## Analysis of Optimization and Generalization Ability

We found SaBN benefits NAS w.r.t. both the generalization ability and the optimization. In DARTS, the goal is to learn the architecture parameter $\alpha$ and the model weights $\omega$ jointly. Specifically, $\alpha$ is optimized to minimize the architecture loss $L_{\text{val}}(\omega, \alpha)$ using validation dataset, where the weights $\omega$ is optimized to minimize the weight loss $L_{\text{train}}(\omega, \alpha)$ with training dataset.

In our experiment, we first visualize the architecture loss as well as weight loss in Fig. 13. Compared with DARTS, the loss value of DARTS-CCBN is larger in the first several epochs, but becomes similar in the later. This indicates that the independent affine layers slow the optimization at the beginning, due to the additional parameters. Compared DARTS-SaBN with DARTS-CCBN, we can observe the losses of DARTS-SaBN is lower than DARTS-CCBN at early stage, indicating the additional sandwich affine layer is beneficial to the optimization of the model. This is achieved by the injected inductive bias of commodity for features from different operations, leading to easier optimization. However, the losses of both DARTS and DARTS-CCBN become lower than DARTS-SaBN in the later stage. This is caused by the architecture collapse in the later searching stage, which can be observed in Fig. 8, 9, where the supernet starts to be dominated by skip connections in DARTS and DARTS-CCBN, leading to a easier optimization Zhou et al. (2020).

We further visualize the gap between the $L_{\text{val}}(\omega, \alpha)$ and $L_{\text{train}}(\omega, \alpha)$ in Fig. 14, with the setting of DARTS, DARTS-CCBN and DARTS-SaBN. The gap in DARTS-SaBN is lower than that of DARTS and DARTS-CCBN, indicating SaBN can also help to improve the generalization ability of the supernet.The inserted shared sandwich affine layer also serves as a regularization term here, since it is updated by data from all previous operations, thus improving the model generalization ability.

## CAPV of SaBN in DARTS

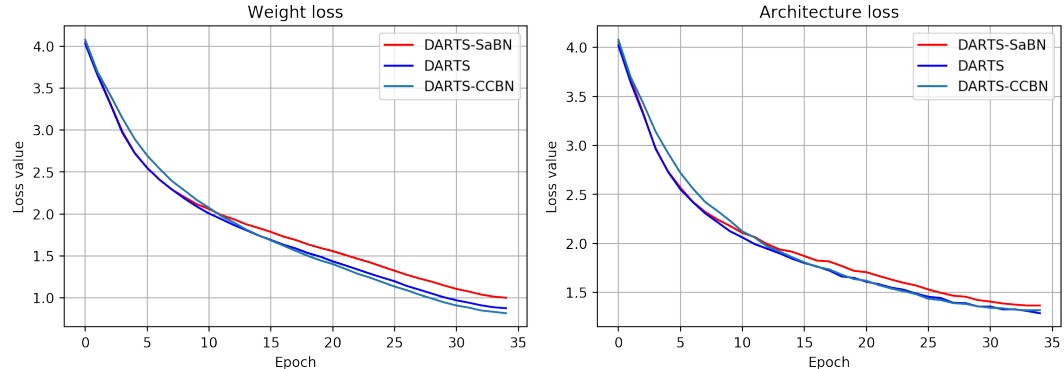

**Figure 13:** The architecture loss $L_{\text{val}}(\omega, \alpha)$ and the weight loss $L_{\text{train}}(\omega, \alpha)$ of DARTS, DARTS-CCBN and DARTS-SaBN.

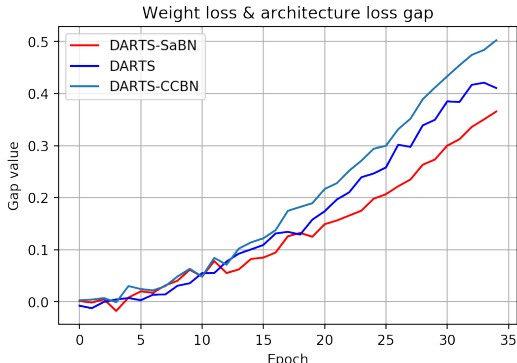

**Figure 14:** The architecture-weight loss gap of DARTS, DARTS-CCBN and DARTS-SaBN.

We use CAPV as the measurement to analyze the behavior of SaBN in DARTS, which is introduced in Sec. 3.1. We compare our proposed DARTS-SaBN with DATRS-CCBN, w.r.t. their CAPV at each layer, in Fig 15.

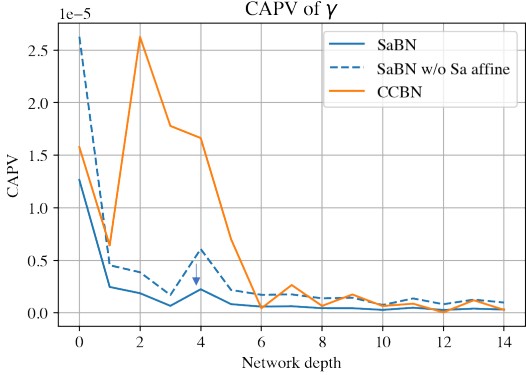

**Figure 15:** The CAPV for DARTS-CCBN and DARTS-SaBN. The shared sandwich affine layer is learned to reduce the feature heterogeneity across the whole network.

### A.2.3  THE RESULTS OF GDAS

GDAS (Dong & Yang, 2019) is an advantaged extension of DARTS (Liu et al., 2018) where the forward activation of all paths during search is replaced by the activation of a sampled single path. This is achieved by introducing Gumbel-softmax to construct a differentiable one-hot vector $[\alpha_0^*, \alpha_1^*, \alpha_2^*, ...\alpha_{n-1}^*], \alpha_i^* \in \{0, 1\}$, from weight vector $[\alpha_0, \alpha_1, \alpha_2, ...\alpha_{n-1}]$.

We conducted four experiments, the original GDAS, GDAS-affine, GDAS-CategoricalCBN and GDAS-SaBN, which are quite similar to DARTS experiments settings. The only difference is that the category index in CCBN and SaBN is obtained by using Gumbel-softmax instead of multinomial sampling. For all experiments we use batch statistics instead of running mean and variance(Dong & Yang, 2020).

**Table 5:** Comparison of four variants of GDAS on NAS-Bench-201. Our GDAS-SaBN achieves the highest top-1 accuracy.

| Method | CIFAR-100 | ImageNet |
|---|---|---|
| GDAS | $67.52 \pm 0.06$ | $39.16 \pm 0.32$ |
| GDAS-affine | $64.79 \pm 5.22$ | $42.57 \pm 1.05$ |
| GDAS-CCBN | $57.62 \pm 12.22$ | $32.52 \pm 11.48$ |
| GDAS-SaBN (ours) | $\mathbf{68.67 \pm 1.02}$ | $\mathbf{43.47 \pm 2.01}$ |

The experiments results are shown in Fig. 16. On CIFAR-100, only GDAS-SaBN outperforms the original GDAS. On ImageNet16-120, Both GDAS-SaBN and GDAS-affine improve the search results, while GDAS-SaBN takes the lead finally. The comparative results of GDAS-SaBN and GDAS-CCBN indicate the importance of the shared sandwich affine layer. The ground-truth accuracy of the final searched architecture is summarized in Tab. 5.

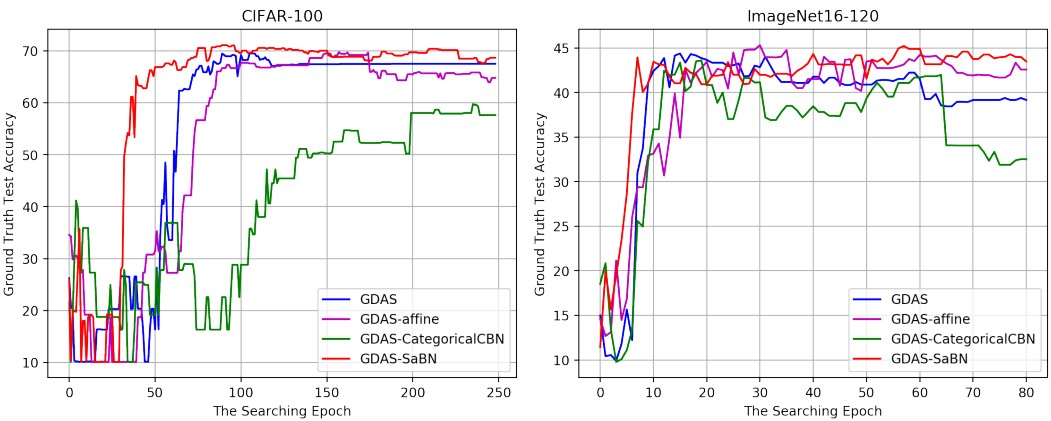

**Figure 16:** Results of architecture search on CIFAR-100 and ImageNet16-120, based on GDAS. Each curve is the average result of three runs using different random seeds.

### A.3    ADDITIONAL RESULTS OF ADVERSARIAL ROBUSTNESS

#### A.3.1    ANALYSIS OF OPTIMIZATION AND GENERALIZATION ABILITY

Our experiments also indicate that SaAuxBN benefits the model on both optimization and generalization. We mainly focus on two losses in our experiments, clean branch loss $L(f_{\text{clean}}(x_{\text{clean}}), y)$ and adversarial branch loss $L(f_{\text{adv}}(x_{\text{adv}}), y)$, where $f$ denotes the model and $x, y$ denotes the input data, label respectively. We visualize the clean branch loss $L(f_{\text{clean}}(x_{\text{clean}}), y)$ and adversarial branch loss $L(f_{\text{adv}}(x_{\text{adv}}), y)$ of SaAuxBN and AuxBN on training set and test set in Fig. 17, 18. The above two figures show that SaAuxBN leads to lower training and testing loss, demonstrating SaAuxBN is beneficial to the model optimization. This is achieved by the injected inductive bias of commodity for features from different domain (clean and adversarial), leading to easier optimization.

Besides, we also show the train-test loss gap of both losses in Fig. 19. For adversarial branch loss $L(f_{\text{adv}}(x_{\text{adv}}), y)$ in the left side, we can observe SaAuxBN has significantly smaller loss gap compared with AuxBN. The train-test loss gaps of clean branch loss $L(f_{\text{clean}}(x_{\text{clean}}), y)$ for SaAuxBN and AuxBN in the right are quite similar. These evidence prove that SaAuxBN also benefits the model in generalization ability. The inserted shared sandwich affine layer also serves as a regularization term, since it is updated by data from both adversarial domain as well as clean domain, thus improving the generalization ability.

#### A.3.2    CAPV OF SABN OF SABN IN ADVERSARIAL ROBUSTNESS

We visualize the CAPV of model with AuxBN and SaAuxBN which is trained under AdvProp framework in Fig. 20.

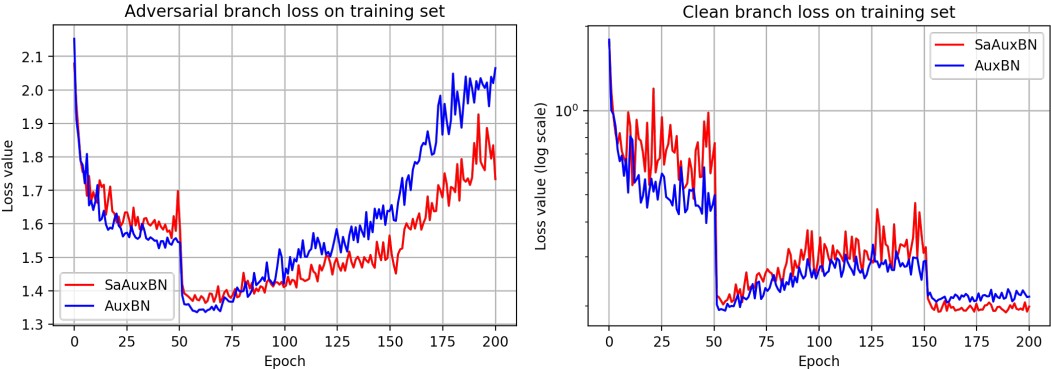

**Figure 17:** The adversarial branch loss $L(f_{\mathrm{adv}}(x_{\mathrm{adv}}), y)$ and clean branch loss $L(f_{\mathrm{clean}}(x_{\mathrm{clean}}), y)$ on training set. $f, x, y$ denote model, input and label respectively. The model with SaAuxBN has lower training loss.

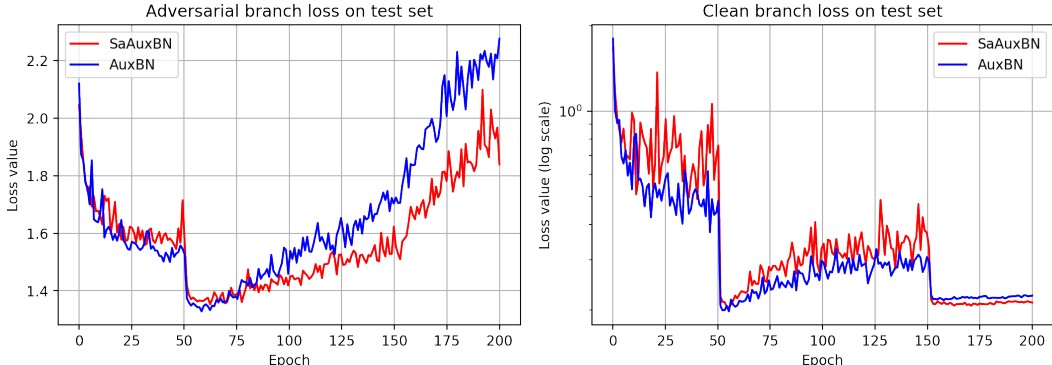

**Figure 18:** The adversarial branch loss $L(f_{\mathrm{adv}}(x_{\mathrm{adv}}), y)$ and clean branch loss $L(f_{\mathrm{clean}}(x_{\mathrm{clean}}), y)$ on testing set. The model with SaAuxBN has lower test loss.

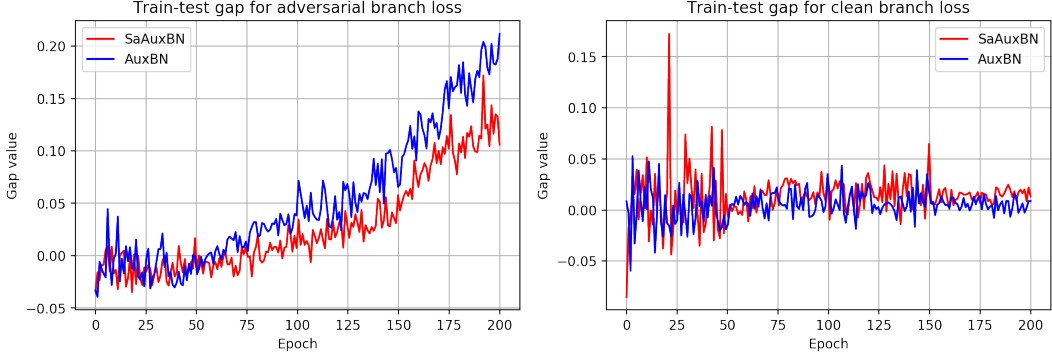

**Figure 19:** The train-test loss gap for adversarial branch loss $L(f_{\mathrm{adv}}(x_{\mathrm{adv}}), y)$ and clean branch loss $L(f_{\mathrm{clean}}(x_{\mathrm{clean}}), y)$.

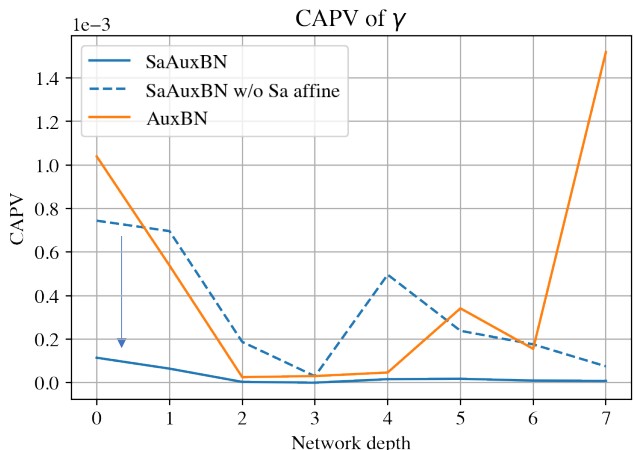

**Figure 20:** The CAPV value for models with AuxBN and SaAuxBN. We can observe the shared sandwich affine layer is learned to reduce CAPV, i.e., the feature heterogeneity.

## A.4 ADDITIONAL RESULTS OF IMAGE GENERATION

### A.4.1 VISUAL RESULTS

**SNGAN Vs SNGAN-SaBN**

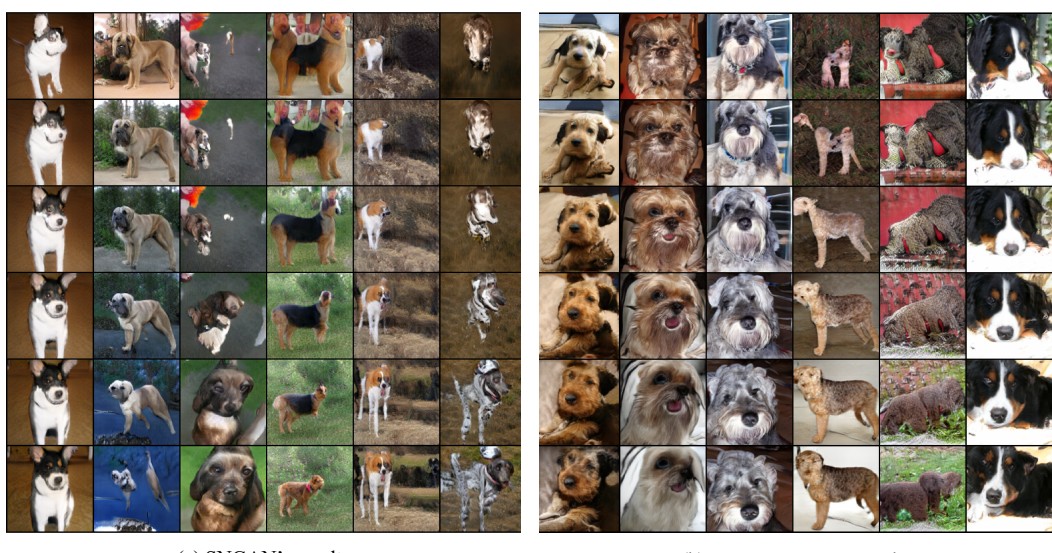

(a) SNGAN's results      (b) SNGAN-SaBN's results

**Figure 21:** The image generation results of SNGAN and SNGAN-SaBN on ImageNet. Each column is corresponding to a specific image class.

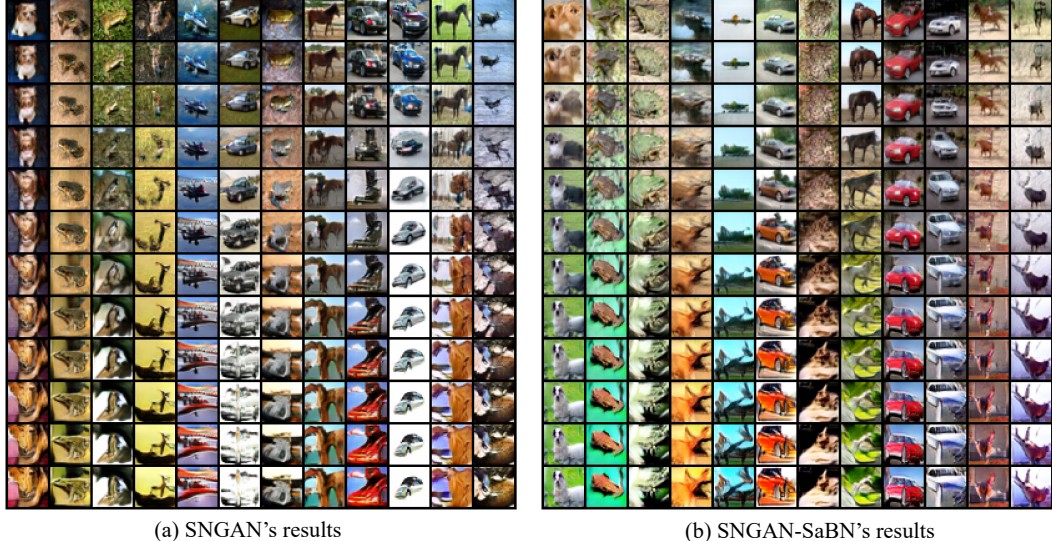

(a) SNGAN's results      (b) SNGAN-SaBN's results

**Figure 22:** The image generation results of SNGAN and SNGAN-SaBN on CIFAR-10.

**AutoGAN Vs AutoGAN-SaBN**

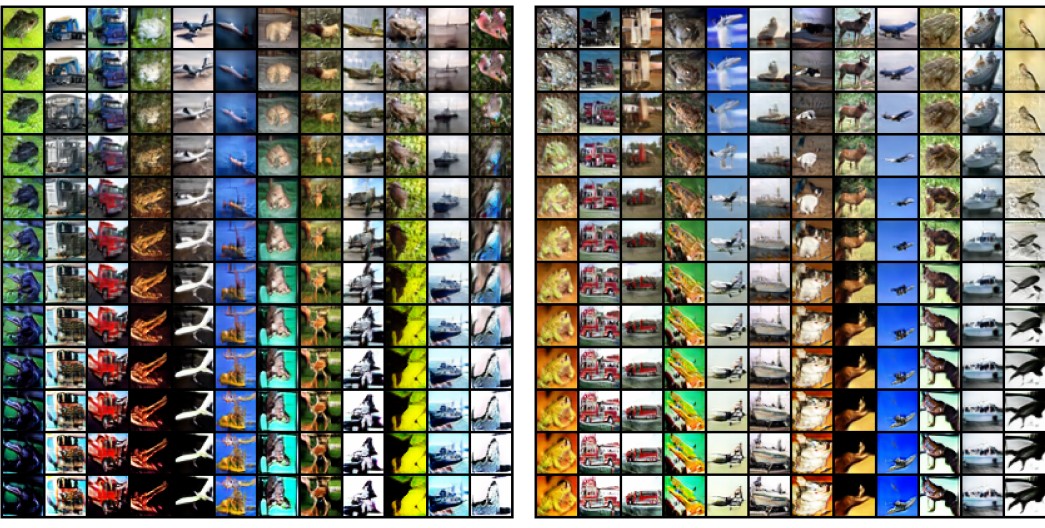

(a) AutoGAN's results        (b) AutoGAN-SaBN's results

**Figure 23:** The image generation results of AutoGAN and AutoGAN-SaBN on CIFAR-10.

**BigGAN Vs BigGAN-SaBN**

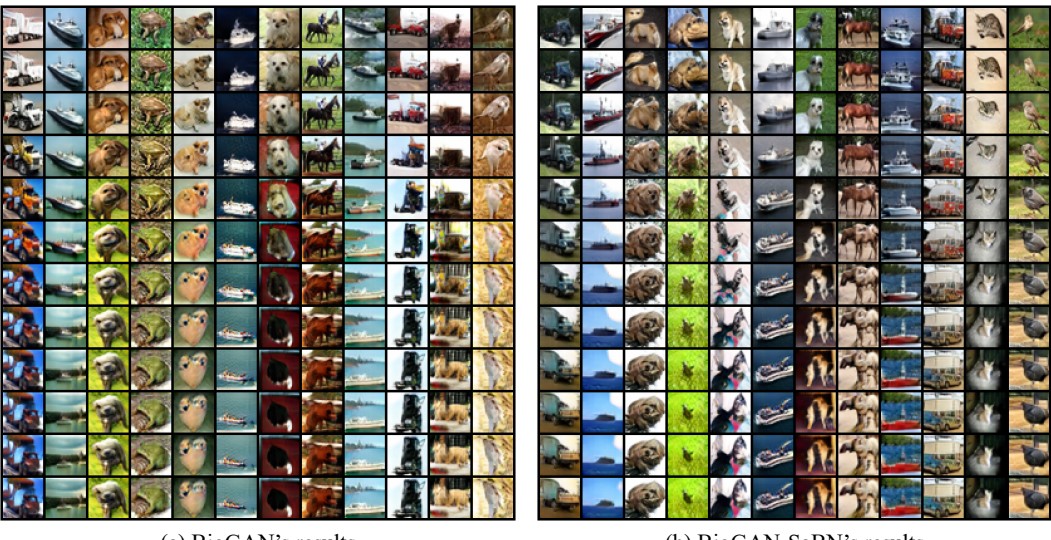

(a) BigGAN's results        (b) BigGAN-SaBN's results

**Figure 24:** The image generation results of BigGAN and BigGAN-SaBN on CIFAR-10.

### A.4.2 ANALYSIS OF OPTIMIZATION

We found that SaBN makes optimization easier. We've visualized the generator training loss of both SNGAN (originally with CCBN) and SNGAN-SaBN on ImageNet dataset in Fig. 25. It can be clearly observed that SNGAN-SaBN has lower generator training loss, yielding better optimization.

### A.5 VISUAL RESULTS OF NEURAL STYLE TRANSFER

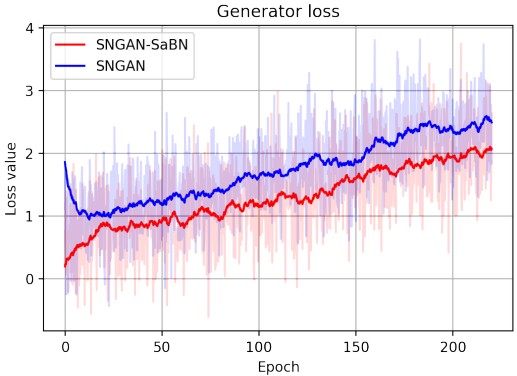

**Figure 25:** The generator loss of SNGAN and SNGAN-SaBN on ImageNet. SNGAN-SaBN achieves lower loss value.

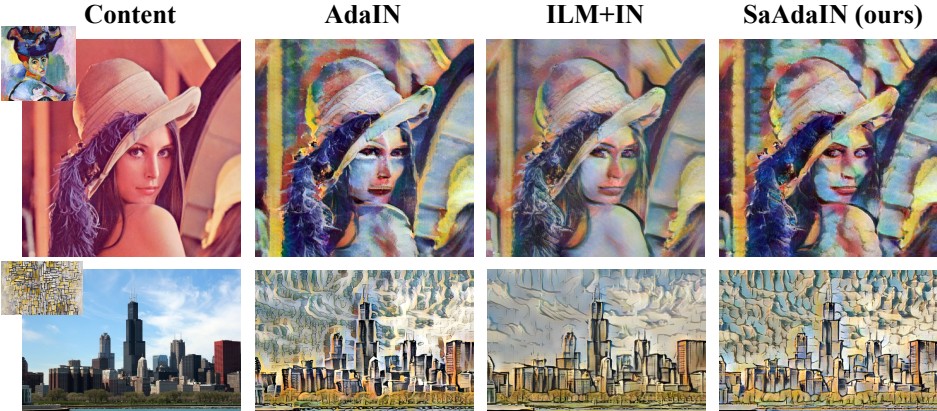

**Figure 26:** The visual results of style transfer. An ideally stylized output should be semantically similar to the content image, while naturally incorporate the style information from the referenced style image.

