# OpenReview forum: "Sandwich Batch Normalization"
_ICLR.cc/2021/Conference — Reject_

### Official Review · AnonReviewer4 · 2020-10-27

**Rating:** 3
**Confidence:** 4

**Review:**

The paper presents a modification to conditional batch normalization, wherein an extra affine layer is introduced between the standardization and the conditional affine layer. The empirical results show the benefits of introducing this extra "sandwich" affine layer.

The intuition behind the approach makes sense, but the formulation makes it difficult to see why SaBN is in fact beneficial compared to CCBN. It does not appear that the approach imposes any restrictions or regularization on the CCBN affine parameters; therefore the only difference between CCBN and SaBN seems to be a different parameterization. I would then expect both CCBN and SaBN to reach the same optimal training loss. It may be that the reparameterization provided by SaBN yields the optimization trajectories that lead to better-generalizing solutions, but it is not explained in the paper whether, or why, this happens. One way to probe this might be to reparameterize each gamma in BN or CCBN as $\gamma=\gamma_1 \gamma_2$ (or even $\gamma = \gamma_1^2$) and study the behavior of the resulting model.

The paper proposes to measure the heterogeneity in the found representations (between different branches of CCBN) via the CAPV measure. In its definition, I could not find what the overbars signify but I assume it means the average over the channels. Also, the indexing of gammas from 0 to N should probably be from 1 to C, for consistency with Eq. (2). The definition of CAPV as the variance of gammas seems problematic, however, in ReLU models with batch normalization: models whose gammas differ by a constant factor represent the same function, so the variance of gammas can be arbitrarily changed without affecting the model. A more useful measure of heterogeneity would need to take this scale invariance into account.

The paper shows several empirical studies, including one for architecture search -- with the main paper using DARTS, and the appendix using GDAS. This choice seems suboptimal, given that in the NAS-Bench-201 paper (https://openreview.net/pdf?id=HJxyZkBKDr), Table 5 seems to indicate that DARTS performs much worse than GDAS. In this work, the appendix devoted to GDAS seems to indicate that (1) there is no consistency on using or not using affine, between CIFAR-100 and Imagenet, and (2) GDAS-SaBN is not statistically significantly better than the better of GDAS and GDAS-affine.

The results in this paper are encouraging, but I believe the paper needs to explain more clearly why SaBN is expected to work (given that it preserves the hypothesis class as well as the minimum of the training objective). Additionally, since it appears to amount to a per-dimension reparameterization, the reader might expect that some of the other reparameterizations could have a similar effect (including such simple interventions as changing the learning rates for some of the gamma or beta parameters), and compellingly demonstrating that the specific reparameterization given by SaBN outperforms such alternatives would make the paper stronger.

---

> ### Author Response · Authors · 2020-11-20
> **Response to Reviewer 4**
>
> We greatly appreciate your time and comments. We would like to re-emphasize our main idea, which was commented by R2 to have ”neat motivation and clear contributions”. The key observation of our paper is that data/model homogeneity and heterogeneity problems widely exist but largely ignored. We demonstrate seemingly different tasks suffer from similar entangled homogeneous/heterogeneous features, and can be solved by our SaBN as a unified and surprisingly simple solution.
>
> 1. *It does not appear that the approach imposes any restrictions or regularization on the CCBN affine parameters.*
>
> Thank you for your great question. In summary, the different designs between our SaBN and CCBN implicitly include the regularization during training. As CCBN is only equipped with independent affine layers, it lacks a global shared transformation induced by all training data, which may lead each independent affine parameter to overfit the subset of training data. However, as our shared affine layer is updated by all training data, each independent affine is further regularized by the global statistics introduced by this shared affine layer. In other words, not only the re-parameterization but also different training strategies of $\gamma_i$ and $\gamma_{sa}$ leads to the advantage of our SaBN.
>
> 2. *The issue of CAPV.*
>
> Thank you for your great points!
> We agree with your point that multiplying a constant factor on gamma won’t change a discriminative function’s prediction result for ReLU networks.
> However, the scalar value $\boldsymbol{\gamma}_i$ here is the averaged value of $\gamma_i$ across all the channels. It is deemed as an approximate statistical measurement of $\gamma_i$ which is only used to measure the discrepancy among different $\gamma$. Therefore, changes in $\boldsymbol{\gamma}_i$ may not be directly interpreted as a uniform scaling over all channels.
>
> Also, thank you for spotting our inconsistency issue and some missed explanation on symbols. We really appreciate your constructive feedback! The correct CAPV definition for CCBN and SaBN have been **updated in our paper**. The overbar indeed denotes the average over the channel dimension, which is also clarified in our updated paper.
>
> 3. *Should place GDAS into the main text & GDAS-SaBN is not significantly better than GDAS.*
>
> Thanks for your suggestions! We will make sure to put the results of GDAS in our paper’s main text instead in our final version.
>
> We further point out that the less SaBN gain on GDAS compared to DART is well expected by us. Compared to DARTS, the main feature of GDAS is to adopt a “hard” single-path style architecture sampling strategy during searching,  disentangling the contribution of each operation, thus enforcing operations to compete more fairly. If you could also please check our response and added analysis to R1 Q2 and R2 Q3, we find SaBN to improve DARTS in a similar taste: it disentangles the model heterogeneity via architecture-dependent independent affines, thus promoting fair competition among the operators. Since GDAS also aims for this same bottleneck, it is understandable that it is already much better at ensuring fair competition compared to vanilla DARTS, there the further gain of SaBN might look smaller. However, we are pleased to see that SaBN can still bring some improvement when applied on top with GDAS, showing more achievable performance room if they are applied together.
>
> 4. *DARTS performs much worse than GDAS in the original NAS-Bench-201 paper [1].*
>
> As stated in our paper, we adopted the early-stopping strategy [2] on our DARTS-based NAS experiments. The searching epoch of DARTS is originally set to 50 epoch in [1], leading to a bad searching result due to the overfitting of the model weights [2], which is confirmed by the author of [1]. In our paper, we empirically set the searching epochs to 35 on both CIFAR-100 and ImageNet16-120 and find it works well for the baseline DARTS.
>
> [1] Dong, Xuanyi, and Yi Yang. "Nas-bench-102: Extending the scope of reproducible neural architecture search." arXiv preprint arXiv:2001.00326 (2020).
>
> [2] Liang, Hanwen, et al. "Darts+: Improved differentiable architecture search with early stopping." arXiv preprint arXiv:1909.06035 (2019).

---

> > ### Comment · AnonReviewer4 · 2020-11-24
> > **Response**
> >
> > Thank you authors for the careful response.
> >
> > I remain unconvinced on points 1 and 2.
> >
> > (1) I do understand that the shared parameters are affected by all the data while the per-branch parameters are optimized on each subset independently. As an intuition, this is reasonable. However, this does not modify the hypothesis class (since for any values of the shared parameters, optimizing the per-branch parameters would give the same loss, albeit with different optimization dynamics).  I appreciate the positive empirical results, but believe that it is important to understand the cause of this improvement, and the explanations given in the paper and in the rebuttals do not help me understand these causes. Echoing R1, I do not know whether the benefit comes from better training and/or better generalization, and why is either one enabled by SaBN.
> >
> > I don't actually think that it is necessary to formally prove results for SaBN training or generalization (though it would of course be nice), but I would at least like to understand how training steps differ (in terms of gradients, updates, etc) between CCBN and SaBN and how these differences affect the training or make SaBN work better in the architecture search context.
> >
> > (2) I may not have fully understood the authors' clarification, but will try to clarify my objection: if in a model with ReLUs and batchnorm we multiply all gammas by the same scalar $s$ (except for the gammas in the model head that are not followed by normalization), this will not change the model outputs, but it will scale CAPV by $s^2$. This suggests that CAPV is not a good way to compare models, since for models that represent the same function we can end up with arbitrarily dissimilar CAPV values.

---

> > > ### Author Response · Authors · 2020-11-25
> > > **Thanks for your further comments.**
> > >
> > > Thanks for your constructive comments.
> > >
> > > 1.  *Why does SaBN benefit training? Better optimization or better generalization?*
> > >
> > > We’ve added experiments and confirmed that the benefit of SaBN comes from both optimization and generalization aspects. In summary, we observe SaBN benefits the optimization in all four tasks.
> > >
> > > **Image Generation.** We found that SaBN makes optimization easier. We’ve visualized the generator training loss of both SNGAN (originally with CCBN) and SNGAN-SaBN on ImageNet dataset in Fig. 25. It can be clearly observed that SNGAN-SaBN has lower generator training loss, yielding better optimization. The shared sandwich affine injects an inductive bias of commodity to the features coming from different classes, thus makes the optimization to be easier.
> > >
> > > **Neural architecture search.** We’ve also found SaBN brings benefit on both optimization and generalization ability, where we include the detailed discussion in Sec. A.2.2.3 (*Analysis of Optimization and Generalization Ability*) of the updated paper.
> > >
> > > **Adversarial robustness.** Our experiments also indicate that SaAuxBN benefits the model on both optimization and generalization, which is concretely discussed in Sec. A.3.1 of the updated paper.
> > >
> > > **Neural style transfer.** Our experiments demonstrate that SaBN is able to help the optimization of the model and improve the generalization ability as well. As stated in the paper, the model is trained with a content loss and style loss. The loss curves of AdaIN and SaAdaIN in Fig. 5 indicates that SaBN can help the model to achieve lower style loss and content loss. This is achieved by the injected inductive bias of commodity for all incoming styles, leading to easier optimization.
> > >
> > > 2. *The issue of CAPV.*
> > >
> > > We fully understand your point w.r.t. CAPV, and we do recognize CAPV is not a flawless metric to measure the discrepancy of different models (paths inside a model), since it would be invalid in the case mentioned in your question. However, we still think that would be a rare corner case that would be extremely hard to meet in our experiments, since its existence must meet **three preconditions**:
> > >
> > > a.  "a model with ReLUs and batchnorm" as mentioned in your question. Therefore the image generation task in our paper can avoid this corner case, since they all contains tanh in their last layers.
> > >
> > > b. The model should be a single-branch network. Taking a residual block as an example, the output of the residual block is $x + f(x)$. By multiplying the gammas in the residual block with a scalar value $a$ (by default, the skip connection doesn't have BN), the output would become $x + af(x)$, which is not a linearly scaling up of the original output. Thus in our case, the experiments of adversarial robustness have adopt ResNet-50 as the backbone, which could avoid the mentioned corner case. The supernet used in our NAS experiments can also avoid the corner case since it originally contains skip connections. Furthermore, the output of each cell is the concatenation of each edge's output, making the linear scaling up to be more impossible.
> > >
> > > c. Even if the above two conditions are met, the corner case will appear unless any independent affine parameter in SaBN satisfy $\gamma_i = k\gamma_j$, where $k$ is a arbitrary constant. This condition is also hard to meet since $\gamma$ is a high dimension vector.
> > >
> > > Therefore, we still think CAPV is a valid proxy metric to use in our experiments.

---

### Official Review · AnonReviewer1 · 2020-10-27
**Review summary**

**Rating:** 5
**Confidence:** 4

**Review:**

The authors propose a new variant of batch normalization. Here are my comments.

1. paper organization.

There are only four sections, and it seems that Section 3 is the main section, which contains both contributions and experiments. I would like the authors to clarify that the contribution is only in the formulation paragraph. It seems to me that the paper organization is poor. There are little descriptions, explanations, and discussions about the proposed method (Eqn 3)

2. the contribution.

If I understand correctly, the only difference between previous work (Eqn 2) and the proposed method (Eqn 3) is the additional parameter gamma-sa/beta-sa. At first glance, the two parameters can be absorbed into gamma_i and beta_i in Eqn 2. (I am saying eqn 2 and eqn 3 are equivalent by some reparameterization.

3. experiments

It seems that the experimental results in table 2 are rather poor. Could you give more explanation on this?

Overall, I don't think the current format, evaluations, and paper significance fit the conference.

---

> ### Author Response · Authors · 2020-11-20
> **Response to Reviewer1 (part1/2): poor paper organization**
>
> We thank you for the comment, but respectfully cannot agree with some of your points.
>
> 1. *Poor paper organization.*
>
> We respectfully disagree with your crique on the paper’s organization, and further argue that this seems to come from misunderstanding what is our paper’s true merit. We hope the above explanation can clarify your confusions or concerns.
> Our paper organization is strongly motivated by our key observation: data/model homogeneity and heterogeneity problems widely exist in many applications but were largely ignored, and a principled framework to handle it is absent. Section 4 is to strongly demonstrate that: (1) the four seemingly different tasks suffer from the common challenges of modeling/balancing homogeneous and heterogeneous features; and (2) that common challenge can be fixed by our SaBN, as a surprisingly simple, elegant, hassle-free and unified solution. The two main points contribute far more insights and merits beyond just the formulation itself (which the reviewer 1 incorrectly treated as our main or only technical story). As confirmed by R2, our work has “neat motivation and clear contributions”.
>
> It is not trivial to spot the common pain point across these different tasks. It is even more challenging to understand how & why each problem benefits from the simple unified SaBN. Actually, we have NEVER seen prior work connecting all those four works and showing them facing the common challenge of feature homogeneity versus heterogeneity. Leveraging this novel and critical insight,  SaBN brings significant performance improvements on all of them, and can be plug-and-play into many different applications.
>
> Based on the above explanation, we hope it is now crystal-clear why Section 3 is organized in its current way: the whole section, including not only the “simple” Section 3.1 technical formulation (we consider “simple” as a nice word and goal, BTW), but also all experiments demonstrating how & why it works across applications, belong to our holistic contribution. We are hence very confident in the current paper structure and believe it effectively conveys the story that we want to share with the research community.
>
> We can also happily divide Section 3 into a few more sections, each application being one, although we sincerely do not feel that necessary - as explained above We invite more comments on what structure would work better to convey our story, and we’re happy to take reasonable suggestions. Meanwhile, we hope our main technical message and true contribution has now been correctly passed to you, regardless of paper structure.

---

> ### Author Response · Authors · 2020-11-20
> **Response to Reviewer1 (part2/2)**
>
>
> 2. *The proposed method in Eq. 2 (SaBN) is equivalent to Eq.3 (CCBN) with some reparameterization.*
> The sandwich affine can be merged into independent affine during inference, which is also mentioned in our paper, but they are **NOT** equivalent during training phase. This re-paramterization during the training stage acts as a very effective regularizer and injects the desirable inductive bias. We will detail below.
>
> The core reason is that different independent affine parameters are specifically updated by different subsets of the training data or sub-paths of the model, whereas the shared affine layer is updated by all data and thus responsible for capturing global feature statistics. This global shared homogeneity cannot be achieved if $\gamma_{sa}$ and $\gamma_i$ are merged during training. Our experiments results also demonstrate their significant difference. Tab. 1 indicates that GANs equipped with SaBN can easily outperform those using CCBN, without any additional tuning or tricks. The searched results in NAS at Tab. 2 and Tab. 5 demonstrate that SaBN can improve the search quality.
>
> We further analyzed architecture parameters during search. We visualize operator distribution over edges during searching on CIFAR-100 **in Fig. 8, 9, 10 of our updated paper**. We find that NAS with vanilla BN and CCBN usually collapse to operators like “none” and “skip_connect”. These operators are known to be more gradient-flow-friendly and thus take the dominance easily [1,2,3]. However, our SaBN takes into consideration both global feature statistics of all previous layer’s operators (via shared affine) and the local feature statistics of each operator (via sampled independent affine). The shared affine injects an **inductive bias of  commodity** to the features coming from different operators. Then the independent affine parameters separately process incoming features dominated by different previous layer’s operators, and thus further calibrates the unbalanced gradients when updating the architecture parameters. This leads to a more fair competition among the operators, and powerful operators like conv1x1 and conv3x3 can be successfully promoted.
>
> We hope the confusion has been addressed by the above reasoning & results, and the reviewer has now seen the real value of our re-parameterization.
>
> 3. *Experiments results in table 2 are rather poor.*
>
> We apologize but we are quite confused on this “poor” claim. As confirmed by R3, our solution “works well in practice”. GDAS is shown to perform the best on NAS-Bench201 based on their paper [4] and our results further achieve state-of-the-art performance benchmark. Tab. 2 shows that our proposed SaBN benefits DARTS on both CIFAR-100 (+27.51%) and ImageNet16-120 (+8.36%). Furthermore, SaBN also improves GDAS on CIFAR-100 (+1.15%) and ImageNet16-120 (+4.31%) in Tab. 5. We welcome more detailed inputs for us to provide more to-the-point feedbacks.
>
> [1] Chu, Xiangxiang, et al. "Fair darts: Eliminating unfair advantages in differentiable architecture search." arXiv preprint arXiv:1911.12126 (2019).
>
> [2] Zela, Arber, et al. "Understanding and robustifying differentiable architecture search." arXiv preprint arXiv:1909.09656 (2019).
>
> [3] Chen, Xiangning, and Cho-Jui Hsieh. "Stabilizing Differentiable Architecture Search via Perturbation-based Regularization." arXiv preprint arXiv:2002.05283 (2020).
>
> [4] Dong, Xuanyi, and Yi Yang. "Nas-bench-102: Extending the scope of reproducible neural architecture search." arXiv preprint arXiv:2001.00326 (2020).

---

> > ### Comment · AnonReviewer1 · 2020-11-24
> > **Thanks for the response**
> >
> > The rebuttal actually has clarified a large part of my concerns due to some misunderstanding of the paper. But I still have concerns regarding whether the reparameterization can really bring benefit and where the gain comes from.
> >
> > We come to a consensus that SaBN and CCBN are equivalent during inference by mering \gamma_i with \gamma_sa. I am still not sure why it benefits training, as it is a simple reparameterization.
> >
> > The first reason maybe is that the benefit may come from better training. Then the natural answer to this question is whether your method achieves low training loss and why such a parameterization helps the optimization of the model.
> >
> > The second reason maybe is that the benefit comes from better generalization, as you said, the regularization. But here the evidence should be more complex, e.g., you need to compare the training-testing gap. Or if you believe that the reparameterization #regularize# the model, the natural answer is to provide the training curve between your model and the baseline model, showing the baseline model achieves lower training loss but higher validation loss.
> >
> > The best way to answer this question is to provide rigorous theoretical analysis and write down the regularization term explicitly.
> >
> > Due to my misunderstanding of the paper, I raise my score to 5.

---

> > > ### Author Response · Authors · 2020-11-25
> > > **Thanks for the follow-up questions**
> > >
> > > Thanks for your follow up questions. We’ve added experiments following your great suggestions and confirmed that the benefit of SaBN comes from both optimization and generalization aspects. In summary, we observe SaBN benefits the optimization in all four tasks.
> > >
> > > **Image Generation.** We found that SaBN makes optimization easier. We’ve visualized the generator training loss of both SNGAN (originally with CCBN) and SNGAN-SaBN on ImageNet dataset in Fig. 25. It can be clearly observed that SNGAN-SaBN has lower generator training loss, yielding better optimization. The shared sandwich affine injects an inductive bias of commodity to the features coming from different classes, thus makes the optimization to be easier.
> > >
> > > **Neural architecture search.** We’ve also found SaBN brings benefit on both optimization and generalization ability, where we include the detailed discussion in Sec. A.2.2.3 (*Analysis of Optimization and Generalization Ability*) of the updated paper.
> > >
> > > **Adversarial robustness.** Our experiments also indicate that SaAuxBN benefits the model on both optimization and generalization, which is concretely discussed in Sec. A.3.1 of the updated paper.
> > >
> > > **Neural style transfer.** Our experiments demonstrate that SaBN is able to help the optimization of the model and improve the generalization ability as well. As stated in the paper, the model is trained with a content loss and style loss. The loss curves of AdaIN and SaAdaIN in Fig. 5 indicates that SaBN can help the model to achieve lower style loss and content loss. This is achieved by the injected inductive bias of commodity for all incoming styles, leading to easier optimization.

---

### Official Review · AnonReviewer3 · 2020-10-29
**Simple idea and solution that work well in practice**

**Rating:** 6
**Confidence:** 5

**Review:**

This paper considers improving the performance of various normalizers by factorizing the affinity operations in normalization layer into on shared affinity operation, as well as several several independent affinity operation, each of which is corresponding to a specific data distribution. The experiments on various tasks (e.g. NAS, GAN, adversarial defense) demonstrate the effectiveness of proposed methods.

Overall, the idea is simple yet effective. It is a good practical way to improve the performance of BN and IN.

My questions are as follows:

(1)The section 3.2 is still unclear to me. The SaBN output multiple tensors, each of which is associated with the candidate operation in the previous layer. How to merge these tensors as the input for the next layer? Just do the summation? In A.2.3, why the number of independent affine parameter becomes n^2 in the case of multiple connected.

(2)What is the difference between the searched architectures by using BN, CCBN and SaBN? Could the authors give some analysis of the impact on the searched architecture when using SaBN?

(3)In Fig.11, the authors show some generated results on ImageNet. I think more results generated from the original methods should be given for the comparison.

(4)I want to know the limitations or some failure results of the proposed method, so that we can have a more comprehensive understanding of the proposed normalizer.

---

> ### Author Response · Authors · 2020-11-20
> **Response to Reviewer3**
>
> We greatly appreciate your questions and suggestions and hope we can address your questions below.
>
> 1. *Section 3.2 (NAS) is still unclear to me.*
>
> We are sorry for the confusion we've made in this section. In summary, vanilla BN in each operator is replaced by an SaBN module, whose independent affine is conditioned on the operator choice(s) from its upstream in-degree connections. We have **updated our Fig. 3** to be more clear.
>
> Taking NAS-Bench-201 as the example (Fig. 3), there are five possible operators on each edge. Therefore, for an operator that has one upstream edge connecting to it (i.e., it accepts one input feature map from this upstream edge), the SaBN in this operator has five independent affine layers. The choice of independent affines is sampled by the probabilities of this incoming edge’s architecture parameters (i.e., softmax of [$\alpha^{k-1}_0, \alpha^{k-1}_1, \cdots, \alpha^{k-1}_4$]).
> If there are two incoming edges (i.e., it accepts a summation of two input feature maps from two upstream edges), the SaBN in this operator has 25 independent affine layers, with the sampling from the joint distribution of two edges’ architecture parameters.
>
> 2. *What is the difference between the searched architectures by using BN, CCBN and SaBN?*
>
> Thank you for your question! In summary, SaBN in NAS disentangles the model heterogeneity via architecture-dependent independent affines, and promotes fair competition among the operators. This fair competition successfully avoids the differentiable search from collapsing into some gradient-flow-friendly but trivial solutions. **We visualized searched architectures (i.e. the repeated stacked cell) in Fig. 11.**
>
> More importantly, **we visualize operator distribution over edges in Fig. 8, 9, 10.** We found that NAS with vanilla BN and CCBN usually collapse to operator “skip_connect”, which is known to be more gradient-flow-friendly and thus take the dominance [1,2,3]. However, our SaBN takes into consideration both global feature statistics of all previous layer’s operators (via shared affine) and the local feature statistics of each operator (via sampled independent affine). The shared affine injects a homogeneous inductive bias to the features coming from different operators. Then the independent affine parameters separately process incoming features which are dominated by different previous layer’s operators, and thus further calibrates the unbalanced gradients when updating the architecture parameters. This leads to a more fair competition among the operators, and powerful operators like conv1x1 and conv3x3 can be successfully promoted.
>
> 3. *Need more visualization results of GANs.*
>
> Thank you for your great suggestion! We have included more generated images in **Fig. 21, 22, 23, 24** for comparison.
>
> 4. *Potential limitation of current work.*
>
> One limitation of our current SaBN would be the extension to unsupervised learning. SaBN so far relies on class labels or upstream categorical choices (NAS) to choose which independent affine layer to activate. An unsupervised scenario for SaBN would be an interesting open question and we are actively investigating it.
>
> [1] Chu, Xiangxiang, et al. "Fair darts: Eliminating unfair advantages in differentiable architecture search." arXiv preprint arXiv:1911.12126 (2019).
>
> [2] Zela, Arber, et al. "Understanding and robustifying differentiable architecture search." arXiv preprint arXiv:1909.09656 (2019).
>
> [3] Chen, Xiangning, and Cho-Jui Hsieh. "Stabilizing Differentiable Architecture Search via Perturbation-based Regularization." arXiv preprint arXiv:2002.05283 (2020).

---

### Official Review · AnonReviewer2 · 2020-10-29
**Simple approach that achieves acceptable results**

**Rating:** 5
**Confidence:** 5

**Review:**

In this paper, the authors propose Sandwich Affine strategy to separate the affine layer in BN into one shared sandwich affine layer, cascaded by several parallel independent affine layers. Such method should well address the inherent feature distribution heterogeneity in many tasks. Following this idea, the SaAuxBN and SaIN have also been introduced in this paper. The extensive experiments demonstrate the effectiveness of such methods in neural architecture search (NAS), image generation, adversarial training, and style transfer.


Pros:
+ Neat motivation;
+ Simple methods with clear contribution;
+ Extensive experiments;

Cons:
- The implementation details in Sec3.2 in not clear. (1) “Specifically, the number of independent affine layers in the SaBN equals to the total number of candidate operation paths of the connected previous layer.” Are you sure it doesn’t equals to the total number of candidate operations in next layer? (2) I think it will be better to add the SaBN in Fig3, and point out the correspondence between the candidate operations and the conditional affine layer. (3)  “The categorical index i of SaBN during searching is obtained by applying a multinomial sampling”. Could you please provide a more detailed explanation ?

- According to Fig.4, SaBN makes the searching process more stable, could you please provide some explanation about this phenomenon? What is the difference of the architecture parameters (i.e. alpha) learned by BN and SaBN?
- Does the author use SaBN to search the architecture, and use BN to fine-tune the searched model? Some implementation details are missing.

- Do the results in Table2 denote the Top-1 accuracy? If yes, please specify in the caption.

- A lot of reference about the normalization methods published in recent years are still missing.


Other Comment:
If we change the order of the Sandwich Affine layer and Conditioned Affine layer, how about the performance in such case? Could you give some analysis about the different about your methods and the above case?

---

> ### Author Response · Authors · 2020-11-20
> **Response to Reviewer 2**
>
> Thank you for your questions! We'd like to clarify below.
>
> 1. *About "the number of independent affine layers in SaBN".*
>
> The core design of SaBN for NAS is to disentangle the mixed model heterogeneity introduced by the upstream layers. In summary, vanilla BN in each operator is replaced by an SaBN module, whose independent affine is conditioned on the operator choice(s) from its upstream in-degree connections, which means the number of cascaded independent affine layers in each SaBN at $layer_k$ is equal to the number of operations (which is equal to 5 in Fig. 3) at $layer_{k-1}$. Thus the statement in our paper is correct. Following your great suggestion, we have also updated our Figure 3 to be more clear.
>
> 2. *About "the categorical index $i$ of SaBN".*
>
> Taking NAS-Bench-201 as the example (Figure 3), there are five possible operators on each edge. Therefore, for an operator that has one upstream edge connecting to it (i.e., it accepts one input feature map from this upstream edge), the SaBN in this operator has five independent affine layers. The choice of independent affines is sampled by the probabilities of this incoming edge’s architecture parameters (i.e., softmax of [$\alpha^{k-1}_0, \alpha^{k-1}_1, \cdots, \alpha^{k-1}_4$]). We have also pointed this out in our updated paper.
>
> 3. *Why does SaBN make NAS stable?  What is the difference of the architecture parameters learnt by BN and SaBN?*
>
> Thank you for your great questions. In general, SaBN in NAS disentangles the model heterogeneity via architecture-dependent independent affines, thus promoting fair competition among the operators. This fair competition successfully avoids the differentiable search from collapsing into some gradient-flow-friendly but trivial solutions (e.g. “none” and “skip_connect” operations).
>
> Specifically, we analyze architecture parameters during search. We visualize operator distribution over edges during searching on CIFAR-100 in **Fig. 8, 9, 10 of our updated paper**. We find that NAS with vanilla BN and CCBN usually collapse to operators like “none” and “skip_connect”. These operators are known to be more gradient-flow-friendly and thus take the dominance easily [1,2,3]. However, our SaBN takes into consideration both global feature statistics of all previous layer’s operators (via shared affine) and the local feature statistics of each operator (via sampled independent affine). The shared affine injects an inductive bias of  commodity to the features coming from different operators. Then the independent affine parameters separately process incoming features dominated by different previous layer’s operators, and thus further calibrates the unbalanced gradients when updating the architecture parameters. This leads to a more fair competition among the operators, and powerful operators like conv1x1 and conv3x3 can be successfully promoted.
>
> 4. *Does the author use SaBN to search the architecture, and use BN to fine-tune the searched model?*
>
> We do not fine-tune our searched model. All our NAS experiments are conducted on NAS-Bench-201[4], which has already provided the ground-truth accuracy of architectures in the search space, thus there is no need to re-train the searched architecture.
>
> 5. *About NAS results in Table 2.*
>
> Yes, it denotes the top-1 accuracy, we have specified it in our paper. Thank you for pointing it out!
>
> 6. *Missing references.*
>
> Thank you for your great suggestion! We further did another literature survey,  included more recent normalization publications, and updated our related work section. We are also open to include more reference if you would kindly suggest.
>
> 7. *What if swap the order of sandwich affine and independent affine layers?*
>
> Thank you for your great question! We follow your suggestion and conduct the experiments on CIFAR-100.
> - DARTS: 44.05 ± 7.47
> - DARTS-swapped-SaBN: 70.695 ± 3.01
> - DARTS-SaBN (ours) 71.56 ± 1.39
>
> We indeed find your proposed swapped variant of our SaBN can also decouple the homogeneous and heterogeneous features in the NAS problem. Your intuition is correct: both the shared affine layer before and after the individual affines can enforce the homogeneous inductive bias. Meanwhile, our original design still performs better and benefits to more stable results. We will include this discussion in the final paper.
>
> [1] Chu, Xiangxiang, et al. "Fair darts: Eliminating unfair advantages in differentiable architecture search." arXiv preprint arXiv:1911.12126 (2019).
>
> [2] Zela, Arber, et al. "Understanding and robustifying differentiable architecture search." arXiv preprint arXiv:1909.09656 (2019).
>
> [3] Chen, Xiangning, and Cho-Jui Hsieh. "Stabilizing Differentiable Architecture Search via Perturbation-based Regularization." arXiv preprint arXiv:2002.05283 (2020).
>
> [4] Dong, Xuanyi, and Yi Yang. "Nas-bench-102: Extending the scope of reproducible neural architecture search." arXiv preprint arXiv:2001.00326 (2020).

---

### Public Comment · ~Lichar_Yuan1 · 2020-11-16
**More details and analysis should be implemented.**

Hi, Thank you for the interesting work.

I have some questions about this paper.

Q1. How you get the gamma and beta in the sandwich layer in different tasks?  The formulation is not clear.

Q2. In Section3.1, the Result on Image generative is incomplete. Fig.11 only visualizes the SNGAN-SaBN (your model) results, but not show other models , such as AutoGAN, BigGAN.

Q3. In Section 3.2, can you give more experimental details and analysis about NAS? What's the searched model you and the Darts get? Is the experiment fair? The result is not convincing.

Q4. In Section 3.3, I cannot find any definition of "improve SA", what is it mean?

The result seems encouraging, I hope more details should be implemented.  Thanks.

---

> ### Author Response · Authors · 2020-11-17
> **Response to Lichar**
>
> Hi Lichar,
> Thanks for your interests for our work. We hope we have addressed your concern below.
>
> A1. Sandwich Batch Normalization (SaBN) is defined in equation (3) in Sec.3 of our paper, which is applicable for all four tasks in our paper. Generally, a SaBN is composed of a normalization layer, a sandwich affine layer, and a set of independent affine layers. I suppose you are asking about how we set the independent affine layers in different tasks.
> * GAN: Similar to Conditional Categorical Batch Normalization (CCBN), the number of independent affine layers is equal to the number of generation image classes.
> * NAS: As mentioned in our paper, each SaBN is placed on each operation. For each SaBN at $layer_k$ in Fig. 3, the number of independent affine layers is equal to the number of unique operations in $layer_{k-1}$, aiming at disentangling the mixed model heterogeneity during the search process.
> * Adversarial Robustness: A SaBN in this task only has two sets of independent affine layers, which is similar to AuxBN. A clean branch for clean samples and an adv branch for adversarial samples.
> * Style Transfer: There’s no real independent affine layers in SaAdaIN, since the rescale factor (independent gamma and beta) is coming from the statistics of input style images.
>
> A2: Thanks for your suggestion, We've added image generation results for AutoGAN and BigGAN in our updated paper.
>
> A3: For neural architecture search experiments in our paper, we adopt NAS-BENCH-201 as our testbed, where we directly use the provided script for searching[1]. Except for different normalization modules in the supernet, we keep all other settings to be exactly identical. No additional trick is added in our experiment. Therefore there’s no hidden details or unfair settings in our NAS experiments.
>
> We also visualize the searched architectures as well as other methods’ searched architectures **in Fig. 11 in our update paper**.
>
> A4: As stated in Sec. 3.3, SA stands for Standard Testing Accuracy, which means the accuracy a model obtains in a clean (non-adversarial) dataset.
>
> [1] NAS-BENCH-201 https://github.com/D-X-Y/AutoDL-Projects

---

### Author Response · Authors · 2020-11-20
**General response**

We would like to thank all reviewers for providing many useful feedbacks. We sincerely apologize for this delayed response, since the author team had been extremely occupied by another deadline just passed.

Below we address all questions raised and provide point-to-point responses. We thank all reviewers for appreciating our novel insight of feature homogeneity versus heterogeneity, simple & easy-to-use fashion, extensive experiments, and effective performance gains on substantially diverse applications.

We find most confusions of our paper arises from some lack of visualization or interpretation in the original submission, which we have fixed and added thoroughly. We have also clarified some misunderstanding of our main merits, e.g., our main novelty is “one normalization solution to four different problems”, which is far beyond a “simple technical formulation”.

We hope our responses, although coming a bit late in this time window (we apologize again), have clarified all confusions and could help reviewers more fairly and positively assess our work. We thank all reviewers’ time again.

---

### Decision · Program_Chairs · 2021-01-07
**Final Decision**

**Decision:**

Reject

**Comment:**

This work proposes a novel reparameterization of batch normalization that is hypothesized to give a better inductive bias for learning several tasks, including neural architecture search, conditional image generation, adversarial robustness and neural style transfer. The reviewers indicate that this is useful and is of interest to the ICLR audience, but they are not satisfied with the analysis offered in the paper. Specifically, the reviewers request that the authors provide a more detailed analysis of why the proposed reparameterization improves results, given that it does not change the expressive power of the model class. Additionally, the reviewers have some concerns about the structure of the paper. I therefore recommend rejecting the paper at this time.